# Fate of methane in canals draining tropical peatlands

Clarice R. Perryman [1] ✉, Jennifer C. Bowen [1], Julie Shahan [1],
Desi Silviani P.A.B [2], Erin Dayanti[3], Yulita Andriyani[2], Adibtya Asyhari [4],
Adi Gangga[4], Nisa Novita [4,5], Gusti Z. Anshari [2,3] & Alison M. Hoyt[1]

Tropical wetlands and freshwaters are major contributors to the growing atmospheric methane ($CH_4$) burden. Extensive peatland drainage has lowered $CH_4$ emissions from peat soils in Southeast Asia, but the canals draining these peatlands may be hotspots of $CH_4$ emissions. Alternatively, $CH_4$ oxidation (consumption) by methanotrophic microorganisms may attenuate emissions. Here, we used laboratory experiments and a synoptic survey of the isotopic composition of $CH_4$ in 34 canals across West Kalimantan, Indonesia to quantify the proportion of $CH_4$ that is consumed and therefore not emitted to the atmosphere. We find that $CH_4$ oxidation mitigates $76.4 \pm 12.0\%$ of potential canal emissions, reducing emissions by ~70 mg $CH_4$ m$^{-2}$ d$^{-1}$. Methane consumption also significantly impacts the stable isotopic fingerprint of canal $CH_4$ emissions. As canals drain over 65% of peatlands in Southeast Asia, our results suggest that $CH_4$ oxidation significantly influences landscape-scale $CH_4$ emissions from these ecosystems.

Wetlands and freshwaters contribute ~30–55% of global $CH_4$ emissions[1], with significant emissions from tropical ecosystems[2-4]. Rising $CH_4$ emissions from tropical wetlands due to temperature and rainfall anomalies have contributed substantially to the growing atmospheric $CH_4$ burden[5-8]. In addition to climate change, ongoing disturbances to tropical wetlands like deforestation, drainage, fertilizer application, and slash and burn agriculture, as well as rewetting and restoration efforts, stand to impact their contribution to the global $CH_4$ budget[9-14]. However, the impact of tropical wetland disturbance on $CH_4$ cycling is not well understood.

In Southeast Asia, wetland disturbance has heavily impacted peatlands, destabilizing the large pool of soil carbon stored in the peat soils of this region[15,16]. Peatlands in Southeast Asia have undergone extensive drainage for oil palm plantations, timber, and other agriculture through the construction of canals[17,18] that lower the water table and therefore lower $CH_4$ emissions from peat soils[19-21]. Instead, the $CH_4$ produced in peat soils is transported into canals via lateral flow[22,23], increasing the relative importance of drainage canals as a source of $CH_4$ emissions[24,25]. Canals can represent over 50% of peatland

$CH_4$ emissions in Southeast Asia[26], but estimates of the magnitude of drainage canal $CH_4$ emissions vary by several orders of magnitude[27,28]. Given that drainage increases the importance of aquatic carbon fluxes from tropical peatlands[29], and the large uncertainty around canal $CH_4$ emissions, greater understanding of the key controls and mechanisms driving canal $CH_4$ emissions is needed to constrain their role in tropical peatland $CH_4$ budgets.

One process that strongly influences freshwater $CH_4$ emissions is microbial oxidation of $CH_4$ to carbon dioxide. In other tropical freshwaters (e.g., rivers, lakes) $CH_4$ oxidation attenuates $CH_4$ emissions by 40 to nearly 100%[23,30-32]. The fraction of $CH_4$ transported into canals from drained peatlands that is oxidized instead of emitted is highly uncertain, as are the factors that mediate $CH_4$ oxidation in drainage canals. For example, both aerobic and anaerobic methanotrophic microbiota are found in tropical freshwaters[32-35]. As variation in canal water depth and discharge can impact dissolved oxygen in canals[36,37], examining the relationship between $CH_4$ oxidation and dissolved oxygen could inform how $CH_4$ oxidation in canal waters may vary over space and time. Constraining the importance of $CH_4$ oxidation in

[1]Department of Earth System Science, Stanford University, Stanford, CA, USA. [2]Department of Soil Science, Tanjungpura University, Pontianak, Indonesia. [3]Magister of Environmental Science, Tanjungpura University, Pontianak, Indonesia. [4]Yayasan Konservasi Alam Nusantara, Jakarta, Indonesia. [5]The Nature Conservancy, Jakarta, Indonesia. ✉e-mail: crperry@stanford.edu

canals draining tropical peatlands is a key step to improving our understanding of the processes controlling $CH_4$ emissions from these ecosystems, especially in the densely drained peatlands of Southeast Asia where canals can have a disproportionate impact on landscape-scale $CH_4$ emissions.

Here, we address how much $CH_4$ transported from drained tropical peatlands into canals is oxidized instead of emitted to the atmosphere. We quantified the percent of $CH_4$ oxidized in 34 canal reaches that drain peat soils under varying land uses across West Kalimantan, Indonesia (Fig. 1, Supplementary Data 1) through shifts in the $\delta^{13}C$ composition of $CH_4$ during incubation experiments of canal waters and from field observations of in situ canal $CH_4$ concentration and $\delta^{13}C$-$CH_4$ (Fig. S1). We find that 47.3-91.3% of $CH_4$ transported into canals from drained peatlands is oxidized instead of emitted. The fraction of $CH_4$ that is oxidized is influenced by factors including dissolved oxygen, vegetation, and canal water depth. Overall, our results suggest that $CH_4$ oxidation substantially attenuates $CH_4$ emissions from canals, and as a result, may be a significant control of landscape-level $CH_4$ emissions from drained peatlands in Southeast Asia.

## Results and Discussion

### $CH_4$ consumption and isotopic fractionation observed during incubations

To confirm $CH_4$ oxidation occurs in drainage canals, we incubated water from 13 canals at in situ dissolved $CH_4$ and oxygen concentrations and measured the change in dissolved $CH_4$ and $\delta^{13}C$-$CH_4$ over time. On average, $53.8 \pm 25.6\%$ of the initial $CH_4$ was consumed over the incubation period (17.6–99.7%) and $\delta^{13}C$-$CH_4$ increased by $19.8 \pm 17.7‰$ (2.1–67.8‰, Fig. 2A, Table S1). The increase in $\delta^{13}C$-$CH_4$ observed across incubated waters confirmed the loss of $CH_4$ was from microbial oxidation, as $CH_4$ oxidation leaves residual $CH_4$ enriched in $^{13}C$[38,39]. Methane oxidation rates were variable across incubated waters (0.03–5.6 $\mu$mol $CH_4$ $L^{-1}$ $d^{-1}$, Table S1) and were strongly influenced by initial $CH_4$ concentration (Fig. S2). Neither $CH_4$ production nor $\delta^{13}C$-$CH_4$ depletion was observed in the canal waters during the incubation period.

From these data we calculated the first empirically derived isotopic fractionation factors for $CH_4$ oxidation[40] ($\alpha_{ox}$) in peat-draining freshwaters. Ecosystem-specific values for $\alpha_{ox}$ are critical to estimating the percent of $CH_4$ that is oxidized rather than emitted from the natural environment[41,42]. Mean $\alpha_{ox}$ was $1.022 \pm 0.009$ across the incubated canal waters (range: 1.002–1.039; Fig. 2B). The range of $\alpha_{ox}$ encompasses past observations from northern and temperate freshwaters incubated under in situ dissolved $CH_4$ and oxygen concentrations and temperature[42,43], as well as results from incubations of soil from subtropical rice paddies[44] ($\alpha_{ox}$ of 1.025–1.033) that are often used in estimates of $CH_4$ oxidation in tropical freshwaters[30,32]. While $CH_4$ oxidation rates varied with initial $CH_4$ concentration, we did not observe a correlation between $\alpha_{ox}$ and initial $CH_4$ concentration, nor $\alpha_{ox}$ and $CH_4$ oxidation rate (Fig. S2). Recent work in temperate lakes identified temperature, pH, and dissolved $O_2$ as potential controls on $\alpha_{ox}$[43]. Of these factors, $\alpha_{ox}$ was only weakly positively correlated with the initial dissolved $O_2$ present in each of the incubated waters ($p = 0.07$). $\alpha_{ox}$ did not vary between surface and bottom waters of the subset of canals sampled at two depths for incubation experiments ($1.024 \pm 0.006$ vs. $1.023 \pm 0.012$, $n = 4$ canals). As we did not find significant environmental correlates of $\alpha_{ox}$, we used the mean value to estimate in situ $CH_4$ oxidation as discussed below.

### Oxidation mitigates the majority of drainage canal $CH_4$ emissions

We find that the majority of $CH_4$ transported into canals from drained tropical peatlands is oxidized instead of emitted to the atmosphere. Using the laboratory-derived $\alpha_{ox}$ values, measurements of in situ canal water $\delta^{13}C$-$CH_4$ from 34 canal reaches (Supplementary Data 1), and measurements of source porewater $\delta^{13}C$-$CH_4$ (Supplementary Data 2), we estimated that $CH_4$ oxidation consumes $76.4 \pm 12.0\%$ of $CH_4$ transported into canals (range: 47.3–91.3%). Considering the standard deviation of $\alpha_{ox}$ shifts the mean percent oxidized by ~10%, ranging from $65.5 \pm 12.5\%$ to $89.3 \pm 8.9\%$ (Figure. S3). Similarly, considering the standard deviation of the porewater source $\delta^{13}C$-$CH_4$ measurements, the mean percent oxidized could range from $68.2 \pm 16.1\%$ to $82.4 \pm 8.9\%$ (Fig. S3). Our estimate of the fraction of $CH_4$ transported into canals that is oxidized instead of emitted is consistent with Somers et al. (2023), who estimated that 70% of $CH_4$ was oxidized in a canal draining a tropical peatland in Brunei using a reactive transport model. These results are also consistent with past work indicating that oxidation consumes ~80% of $CH_4$ in blackwater rivers in the Amazon[32] that have low pH and high concentrations of aromatic-rich, humic-like dissolved organic carbon like the canals in our study region[29,45]. Compared to previous work in lotic systems that use a similar approach as employed in our study, we find that oxidation mitigates a higher proportion of potential $CH_4$ emissions in drainage canals than in headwater streams in temperate forests[46] ($55.6 \pm 2.8\%$) or boreal peatlands[47] (~60%).

Canal water dissolved $CH_4$ concentration and $\delta^{13}C$-$CH_4$ across our study region supports our finding that $CH_4$ oxidation limits $CH_4$ release from canals. Dissolved $CH_4$ concentration and $\delta^{13}C$-$CH_4$ in canal waters ranged from 0.05 to 31.6 $\mu$M and $-71.9$ to $-34.1‰$, respectively (Fig. 3B), and dissolved $CH_4$ decreased with increasing $\delta^{13}C$-$CH_4$ ($R^2 = 0.43$, $p < 0.001$, Fig. S4). Previous observations in tropical river networks[32] also observed a negative relationship between the concentration of $CH_4$ in river waters and $\delta^{13}C$-$CH_4$. In these rivers $\delta^{13}C$-$CH_4$ also had a positive relationship with gene markers for methanotrophic bacteria, indicating that variation in $CH_4$ concentration and $\delta^{13}C$-$CH_4$ is influenced by $CH_4$ oxidation. The consistent relationship between $CH_4$ concentration and $\delta^{13}C$-$CH_4$ observed across the drainage canals in our study and these tropical rivers supports the idea that differences in dissolved $CH_4$ concentrations between canal reaches are influenced by $CH_4$ oxidation.

It is unlikely that $CH_4$ concentration in canal waters is dictated only by the amount of $CH_4$ originally transported into canals from the surrounding landscape, including $CH_4$ produced in peat soils and canal sediments. Methane produced in ombrotrophic tropical peat soils is highly depleted in $^{13}C$[23,48]. Unlike in lakes where $\delta^{13}C$-$CH_4$ in littoral sediments and adjacent groundwater can differ by more than 10‰[49], porewater $\delta^{13}C$-$CH_4$ has not been shown to differ between canal bottoms and adjacent peat soils[22]. Porewater $\delta^{13}C$-$CH_4$ collected from 6 profiles (40 to 150 cm depth) located alongside canal waters in our study region had a mean $\delta^{13}C$-$CH_4$ of $-85.0 \pm 5.9‰$, which was consistently more depleted than any observed canal $\delta^{13}C$-$CH_4$ value (Supplementary Data 1, Supplementary Data 2). Porewater $\delta^{13}C$-$CH_4$ varied more between sample depths within each profile than between profiles collected across the landscape, suggesting source $\delta^{13}C$-$CH_4$ is similarly depleted in $^{13}C$ throughout the study region. Methane production in the water column could also influence canal water $CH_4$ concentration and $\delta^{13}C$-$CH_4$. However, this is unlikely to explain our results because we did not observe net $CH_4$ production in any of the laboratory incubations of canal waters, as $CH_4$ concentration decreased and $\delta^{13}C$-$CH_4$ increased in all incubated waters (Fig. 2A, Table S1). If canal water $CH_4$ concentration were influenced solely by the total amount of $CH_4$ produced and then transported into canal waters, we would expect canal water $\delta^{13}C$-$CH_4$ to be similarly depleted across canals and not vary systematically with dissolved $CH_4$ concentration. Given that $CH_4$ concentrations varied ~600-fold alongside a ~40‰ range in $\delta^{13}C$-$CH_4$, our results indicate that $CH_4$ oxidation has a significant influence on canal water $CH_4$ concentration and $\delta^{13}C$-$CH_4$.

As $CH_4$ oxidation was a major control of canal water $CH_4$ concentration, diffusive $CH_4$ emissions were also strongly influenced by the percent of $CH_4$ oxidized. Diffusive $CH_4$ emissions estimated from

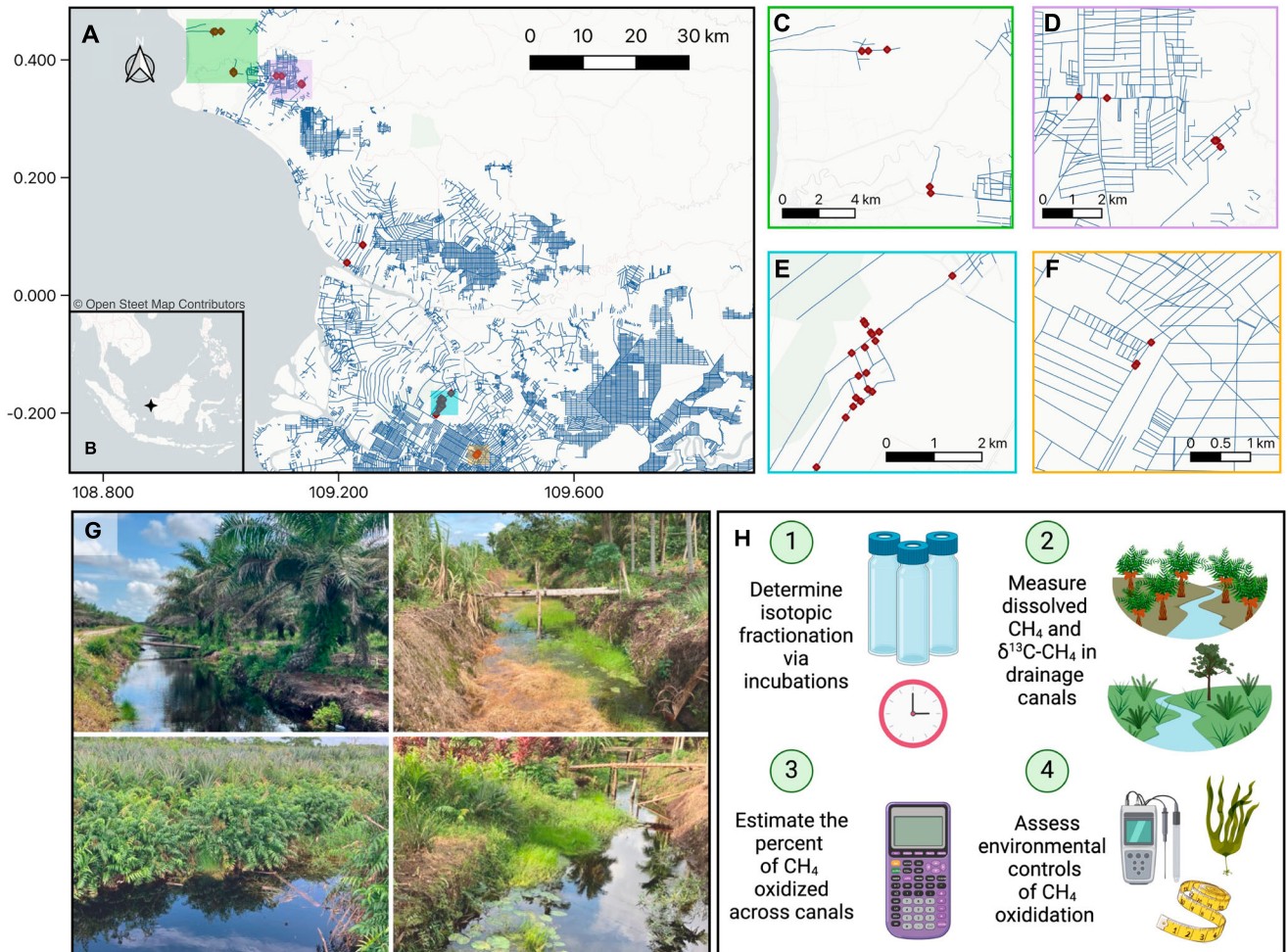

**Fig. 1 | Drainage canal waters were collected in West Kalimantan, Indonesia to measure methane oxidation. A** Study area with drainage canals shown as dark blue lines. **B** shows location of study within insular Southeast Asia. Canal sample locations marked in red points. **C**–**F** Zoomed in view of green, purple, teal, and orange boxes in panel **A**, respectively, showing sample locations. The base map layers in Panels **A**–**F** are available from OpenStreetMap (openstreetmap.org/copyright), available under the Open Database License. **G** Examples of canals in varying land use context and canals with and without aquatic vegetation. **H** Overview of study methods to estimate $CH_4$ oxidation in drainage canals. Created in BioRender. Perryman, C. (2024) BioRender.com/c12r203.

dissolved $CH_4$ concentration (Supplemental Text 1) ranged from 1.0 to 761.8 mg $CH_4$ m$^{-2}$ d$^{-1}$ (mean = 72.2 ± 151.2; median = 18.0) and decreased as the percent of $CH_4$ oxidized increased ($R^2 = 0.54$, $p < 0.001$; Fig. 3D). Measurements of $CH_4$ emissions from floating chamber deployments at a subset of study sites ($n = 12$ canals, mean = 94.9 ± 142.3 $CH_4$ m$^{-2}$ d$^{-1}$, median = 33.0, Supplementary Data 1) also indicated a negative relationship between $CH_4$ emissions and the percent of $CH_4$ oxidized (Fig. S5). By back-calculating what diffusive $CH_4$ flux would be in the absence of oxidation, we estimate that $CH_4$ oxidation reduces drainage canal $CH_4$ emissions by a mean of 136.8 ± 154.1 mg $CH_4$ m$^{-2}$ d$^{-1}$ (range: 9.9–684.2). Given the skewed distribution of dissolved $CH_4$ concentrations that underlie this estimate, the median value of 72.1 mg $CH_4$ m$^{-2}$ d$^{-1}$ (IQR: 36.2–173.6) may be a more robust estimate of the emissions attenuated by $CH_4$ oxidation. Overall, our results provide evidence suggesting that $CH_4$ oxidation mitigates the majority of potential $CH_4$ emissions from canals on the landscape.

### Controls on $CH_4$ oxidation in drainage canals

Of the studied controls on $CH_4$ oxidation, dissolved oxygen and aquatic vegetation had the most significant influence on the percent of $CH_4$ oxidized in canals as determined by canal water $\delta^{13}$C-$CH_4$. We found that the percent of $CH_4$ oxidized increased and dissolved $CH_4$ concentration decreased with the concentration of dissolved oxygen at the canal water surface (0-10 cm; $p < 0.05$, Fig. 4A, Table S2). The relationship between dissolved oxygen and $CH_4$ oxidation is consistent with oxidation mediated by aerobic methanotrophic bacteria, as has been observed in other stream and river networks[32,46]. While all canals had low dissolved oxygen (0.2 to 2.3 mg L$^{-1}$), methanotrophic bacteria of the order Methylococcales have been shown to have the genetic potential for survival and methanotrophic activity in low oxygen environments[50]. Abundant Methylococcales have been identified in hypoxic tropical freshwaters where paired measurements of dissolved $CH_4$ concentration and $\delta^{13}$C-$CH_4$ indicate ongoing $CH_4$ oxidation[34,35]. Our results further support the idea that aerobic $CH_4$ oxidation occurs in tropical freshwaters with low dissolved oxygen.

Furthermore, the percent of $CH_4$ oxidized was higher in vegetated canals than those with open water ($p = 0.01$, Fig. 4B). Vegetation may enhance $CH_4$ oxidation via radial oxygen loss from roots[51,52] or via oxidation by epiphytic methanotrophs in submersed plants[53]. Although we did not observe a significant difference in dissolved oxygen based on the presence of aquatic vegetation ($p > 0.05$, Table S3), oxygen delivered to the water column by aquatic vegetation is likely rapidly consumed by methanotrophs or by competing aerobic heterotrophs as deposition of more labile organic carbon by aquatic vegetation could stimulate heterotrophic respiration in canal waters[29].

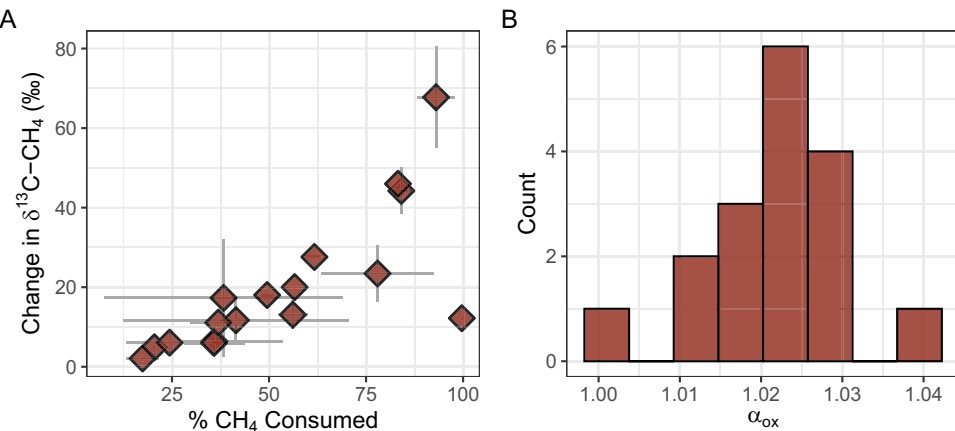

**Fig. 2 | Methane consumption and resulting stable isotope fractionation in incubated canal waters. A** Across incubated waters, $\delta^{13}C$-$CH_4$ increased as the percent of initial $CH_4$ consumed increased. Each data point shows the mean change over ~50 hours of incubation ± standard error of replicates (Table S1). **B** Histogram of $\alpha_{ox}$ values calculated from incubation data. Source data are provided as a Source Data file.

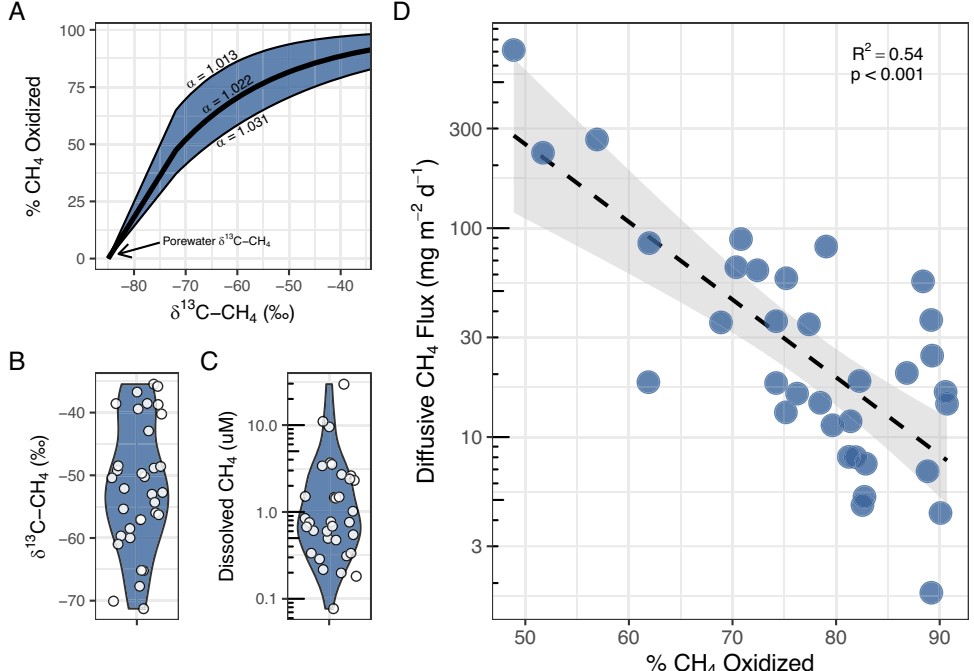

**Fig. 3 | Survey of drainage canal $CH_4$ concentrations and $\delta^{13}C$-$CH_4$ reveal the impact of $CH_4$ oxidation on canal $CH_4$ emissions. A** Curve showing the relationship between canal water $\delta^{13}C$-$CH_4$ and estimated percent $CH_4$ oxidized across the mean (black line) and ± 1 standard deviation (shaded region) of the laboratory derived $\alpha_{ox}$ value. **B, C** Surface water $\delta^{13}C$-$CH_4$ and dissolved $CH_4$ concentration across the studied canals ($n = 34$). **D** Estimates of the percent of $CH_4$ oxidized versus estimated diffusive $CH_4$ flux across the studied canals. For panels **B**–**D** each dot represents a canal. The shaded region of panel **D** represents the 95% confidence interval associated with the linear relationship. Dissolved $CH_4$ concentration and estimated diffusive $CH_4$ flux are shown on a $\log_{10}$ scale in panels **C** and **D**. Source data are provided as a Source Data file.

Lower $CH_4$ concentrations and more enriched $\delta^{13}C$-$CH_4$ in vegetated canals could alternatively be explained by plant-mediated emissions[54], which reduce $CH_4$ concentration and enrich the $\delta^{13}C$ of residual $CH_4$ due to the isotopic fractionation of plant-mediated transport[55]. The deposition of labile organic matter from vegetation could also stimulate acetoclastic methanogenesis, which like $CH_4$ oxidation would contribute towards larger $\delta^{13}C$-$CH_4$ in vegetated canals[39]. However, acetoclastic methanogenesis likely contributes little to the $\delta^{13}C$-$CH_4$ in vegetated canals because hydrogenotrophic methanogenesis has been identified as the dominant pathway in the ombrotrophic tropical peatlands of Southeast Asia[23] and the Americas[48,56]. Disturbance in peatlands in Southeast Asia has been observed to increase the

abundance of plant functional types associated with acetoclastic methanogenesis, like graminoids, but this shift does not appear to increase the abundance of acetoclastic methanogens[57]. While we cannot rule out the possible influence of acetoclastic methanogenesis on canal water $\delta^{13}C$-$CH_4$, the lower dissolved $CH_4$ concentration in vegetated canals ($p = 0.02$, Table S3) lends more support to the idea that vegetation enhances $CH_4$ oxidation rather than acetoclastic $CH_4$ production in canals.

Given that higher dissolved oxygen and the presence of aquatic vegetation were observed in canals with a shallower water depth (Fig. S6), canal water depth may indirectly mediate $CH_4$ oxidation in drainage canal waters. Overall, dissolved oxygen in the surface water of

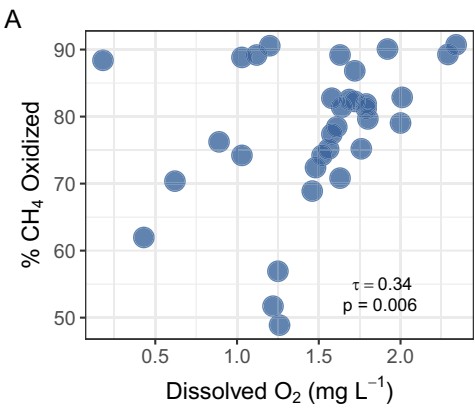

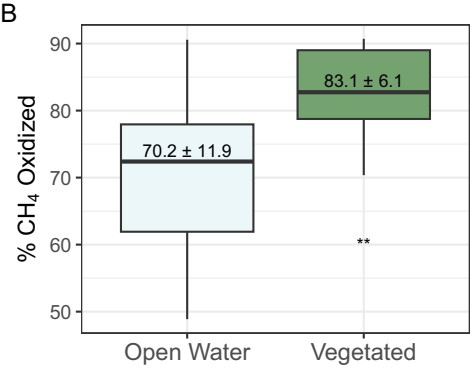

**Fig. 4 | Controls on CH$_4$ oxidation in drainage canals. A** Dissolved oxygen in the surface waters (0–10 cm) of drainage canals versus the percent of CH$_4$ oxidized. Each point represents a canal (*n* = 34). **B** Boxplot of the percent of CH$_4$ oxidized in open water (light blue, *n* = 19) and vegetated (green, *n* = 15) canals. Within each box the black lines represent median values and the height of the boxes represent the interquartile range. Error bars extend up to 1.5 times the interquartile range. The number in each box represents the mean ± 1 standard deviation of the percent of CH$_4$ oxidized for each group. Source data are provided as a Source Data file.

canals (0–10 cm) decreased with the depth of water present in the canal (Kendall's **τ** = −0.41, *p* < 0.05, Fig. S6). Dissolved CH$_4$ concentration, and therefore estimated diffusive emissions, also had a weak but significant positive correlation with canal water depth (**τ** = 0.26, *p* = 0.03, Table S2). This result contradicts previous findings in drainage ditches in temperate peatlands where CH$_4$ emissions had a weak negative correlation with depth[58], but these differing results may be explained by how well canal waters are mixed and aerated. For example, while we observed CH$_4$ oxidation in canals where dissolved oxygen is low (<2.5 mg L$^{-1}$) at the surface, dissolved oxygen may become depleted at depth[29,45] to below the concentration needed for aerobic methanotrophs with high oxygen affinity. As such, CH$_4$ oxidation may be limited to the surface waters of deeper canals, while in shallower canals oxidation may occur throughout the water column. Our study also only explicitly considered diffusive emissions. Measurements of CH$_4$ ebullition from canals could further clarify the role of water depth in shaping net canal CH$_4$ emissions, as ebullitive emissions vary with water depth[59]. Altogether, our results suggest that shallower, vegetated canals may attenuate a higher percentage of CH$_4$ emissions through CH$_4$ oxidation.

Land use and seasonal precipitation cycles can both influence canal water depth and therefore dissolved oxygen. While we did not observe a significant impact of peatland land use on CH$_4$ oxidation nor other parameters including dissolved oxygen (Table S4), peatland water table, which directly influences canal water levels, has been shown to vary significantly between land use types[60]. Canal water depth also varies 2- to 5-fold throughout the year in response to precipitation (Fig. S7), and reduced precipitation and flow during drier months may facilitate oxygen depletion by limiting turbulent mixing and re-aeration of canal waters[27,61]. Accordingly, past studies have reported higher canal CH$_4$ emissions during dry periods[27,28]. While our study was not conducted during pronounced wet or dry periods, the dissolved CH$_4$ and oxygen concentrations measured in our study fall within the range observed across Southeast Asia under varying land uses and seasons[22,28,45,62,63] (Table S5). As such, we anticipate that water column CH$_4$ oxidation is prevalent across canals draining degraded peatlands in Southeast Asia.

### Influence of oxidation on CH$_4$ emissions and their $^{13}$C in drained tropical peatlands

Our observations of canal CH$_4$ emissions estimated from dissolved CH$_4$ concentration (72.2 ± 151.2 mg CH$_4$ m$^{-2}$ d$^{-1}$) and collected using floating chambers (94.9 ± 142.3 mg CH$_4$ m$^{-2}$ d$^{-1}$) are within range of past observations from Indonesia[27,28,64] and Malaysia[26] where mean

emissions range from 2.8 to 1073 mg CH$_4$ m$^{-2}$ d$^{-1}$ (Table S6). The IPCC CH$_4$ Emissions Factor for canals in tropical peatlands of 618.9 mg CH$_4$ m$^{-2}$ d$^{-1}$ (2259 kg CH$_4$ ha$^{-1}$ y$^{-1}$) was based on the only reported data[27] at the time of the 2013 Wetlands Supplement[65] This emission factor now represents the high end of field estimates to date among a still small number of existing studies and should be reconsidered to more accurately inventory the anthropogenic (e.g., from land use change) component[66] of CH$_4$ emissions from degraded tropical peatlands.

Despite high oxidation efficiencies, drainage canals can still emit large amounts of CH$_4$. For example, in canals where ~50% of the CH$_4$ transported from peatlands is oxidized we observe emissions >200 mg CH$_4$ m$^{-2}$ d$^{-1}$ (Figs. 3, S5). The canals in this study were primarily situated in smallholder agricultural systems (Supplementary Data 1), and the mean estimated diffusive CH$_4$ emissions from canals presented here are 30x larger on a per area basis than mean peat soil CH$_4$ emissions from smallholder agriculture fields in West Kalimantan[67]. Thus, while CH$_4$ oxidation plays a critical role in attenuating canal CH$_4$ emissions, canals can still contribute significantly to landscape-level CH$_4$ emissions from drained peatlands in Southeast Asia.

Beyond the rate of emissions, the $\delta^{13}$C signature of CH$_4$ emitted from tropical wetlands and freshwaters are critical for constraining their contribution to the global CH$_4$ budget, as $\delta^{13}$C-CH$_4$ values underpin source partitioning by atmospheric inversion models. Using a floating chamber to capture CH$_4$ emitted from a subset of the studied canals, we found that the mean $\delta^{13}$C-CH$_4$ was -64.7 ± 10.5‰ (Fig. 5A). Canal CH$_4$ emissions generally decreased as emitted $\delta^{13}$C-CH$_4$ increased (Fig. S8), therefore the flux-weighted mean $\delta^{13}$C-CH$_4$ was more negative at -69.0 ± 5.7‰ (Fig. 5A). Past observations of the $\delta^{13}$C signature of tropical wetland CH$_4$ emissions[68] indicate a range of -64‰ to -53‰. Our results suggest that the $\delta^{13}$C signature of CH$_4$ emissions from drainage canals, and potentially drained peatlands in Southeast Asia as whole due to the contribution of canals to landscape-scale CH$_4$ emissions, is more negative than prior measurements from tropical wetlands. As such, implementing a distinct $\delta^{13}$C-CH$_4$ source signature for Southeast Asian peatlands may improve top-down estimates of their CH$_4$ emissions.

Furthermore, we find that the variation in $\delta^{13}$C of CH$_4$ emissions from the canal water surface (−86.9 to -44.3‰) was largely explained by the percent of CH$_4$ oxidized in canal waters ($R^2$ = 0.68, *p* < 0.05; Fig. 5B). Previous studies have identified oxidation, alongside variation in methanogenic pathways and wetland vegetation, as one potential explanation for latitudinal differences in the $\delta^{13}$C of wetland CH$_4$ emissions[68–70]. Our results indicate that once CH$_4$ produced in peat soils is transported into canals, both the magnitude and the isotopic

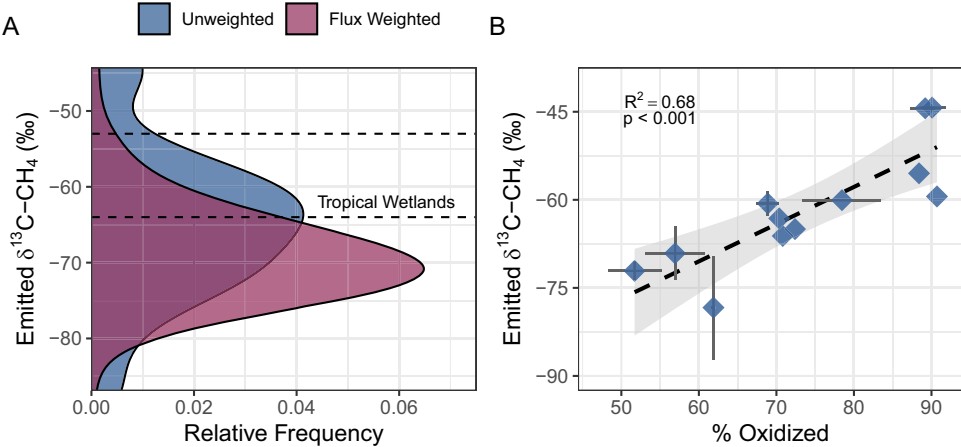

**Fig. 5 | The isotopic composition of CH$_4$ emissions from tropical peatland drainage canals. A** Probability density estimates of the δ$^{13}$C of CH$_4$ emitted from canal waters showing the unweighted (blue) and flux-weighted (purple) distributions of emitted δ$^{13}$C-CH$_4$. The dashed black lines show the range of δ$^{13}$C of tropical wetland CH$_4$ emissions reported in ref. 68. **B** The percent of CH$_4$ oxidized in drainage canal waters (estimated from dissolved δ$^{13}$C-CH$_4$ using α$_{ox}$ = 1.022) versus the

δ$^{13}$C of CH$_4$ emitted from the corresponding canal. Each point represents a canal ($n$ = 12) and error bars show the mean ± 1 standard deviation if replicates were collected at a canal. The shaded regions represent the 95% confidence interval associated with the linear relationship shown in panel B. Source data are provided as a Source Data file.

signature of canal CH$_4$ emissions are strongly influenced by CH$_4$ oxidation.

In summary, we demonstrate that CH$_4$ oxidation can substantially attenuate CH$_4$ emissions from canals draining peatlands in Southeast Asia. We estimate that CH$_4$ oxidation mitigates >50% of potential CH$_4$ emissions from canals across West Kalimantan, Indonesia. As landscape-scale measurements of CH$_4$ exchange in drained tropical peatlands indicate that canal networks contribute disproportionately to emissions from these ecosystems[20], our results suggest that CH$_4$ oxidation influences emissions not only from drainage canals but from degraded peatlands in Southeast Asia as a whole. Our results also have implications for peatland CH$_4$ emissions in response to land use change, including peatland restoration efforts. For example, we find that oxidation attenuates more CH$_4$ emissions from shallower canals that have higher dissolved oxygen concentrations. As such, efforts to rewet drained peatlands in Southeast Asia through canal blocking may impact CH$_4$ oxidation and therefore canal CH$_4$ emissions through changing canal water depth. Given the extensive networks of drainage canals in Southeast Asia and their substantial contribution to peatland CH$_4$ emissions, land use changes impacting CH$_4$ oxidation in canals will be reflected in the contribution of peatlands in Southeast Asia to the global CH$_4$ budget.

## Methods
### Field sampling
Drainage canals in lowland peatlands were sampled in Kubu Raya and Mempawah Districts, West Kalimantan, Indonesia. Canals were sampled in Kubu Raya in May 2023 and Mempawah in April 2024. This region has an equatorial rainfall pattern with no clear wet and dry season[71]. There is heavy rainfall year-round, but the driest months of the year usually occur in July or August. We sampled waters from canals of different sizes (5 to 90 cm water depths, 0.5 to 6 m canal widths), canals with ($n$ = 15) and without aquatic vegetation ($n$ = 19), and canals situated on peatlands under a variety of land uses. Smallholder mixed agriculture is the most represented land use in this study, but the sampled canals also include areas in smallholder plantations (pineapple and oil palm), industrial oil palm plantations, and open undeveloped land (i.e., deforested and/or burned areas), as well as 1 canal in a degraded forest, to capture the heterogeneity of drainage canals in the region. At each canal, we measured the canal dimensions as well as water temperature (°C), pH, dissolved oxygen (mg L$^{-1}$),

conductivity (μS cm$^{-1}$), and redox potential (Eh, in mV) using a Hanna Instruments HI9829 multiparameter meter. A summary of the canals included in this study is available in Supplementary Data 1.

To measure the isotopic composition of source CH$_4$, we collected porewater profiles at 6 locations adjacent to a subset of the sampled canals. As shallow porewater is the primary source of discharge to drainage canals[22], porewater was collected from 4-5 depths between 40 cm and 150 cm pending water table depth. Porewater was collected using a portable piezometer made of 3/8" stainless steel tubing housing 1/4" polyethylene tubing equipped with a coarse polypropylene screen to prevent collection of coarse debris (SedPoints, M.H.E. Products). Porewater samples were stored in 12 mL glass Exetainer™ vials (Labco Ltd.) without headspace and acidified in the field to a pH of less than 2 using 1.5 M HCl.

### Canal CH$_4$ concentration and δ$^{13}$C
We collected surface water samples for analysis of dissolved CH$_4$ concentration and δ$^{13}$C-CH$_4$ at all canals. Canal water samples were collected approximately 5 cm below the water surface and stored in 12 mL glass Exetainer™ vials (Labco Ltd.) without headspace. Canal waters collected for assessment of in situ dissolved CH$_4$ concentration and δ$^{13}$C-CH$_4$ were acidified in the field to a pH of less than 2 using 1.5 M HCl. Canal water samples collected in 2023 (including incubations described below) and all porewater samples were analyzed at the Stable Isotope Facility at UC Davis via a Delta V Plus IRMS following headspace equilibration. Samples collected in 2024 were analyzed at Stanford University via a Picarro G2210-$i$ cavity ring down spectrometer following headspace equilibration. Reference standards with CH$_4$ mixing ratios and δ$^{13}$C-CH$_4$ of 10 ppm/-45.5‰ and 30 ppm/-69.0‰ were run before and after sample analysis on the Picarro G2210-$i$ to check for accuracy and instrument drift. Dissolved CH$_4$ concentrations were calculated considering the mixing ratio of CH$_4$ in the equilibrated headspace, using the ideal gas law, and in solution, following Henry's Law, using the neonDissGas package[72].

### Incubations
We collected canal waters at a subset ($n$ = 13) of the drainage canals for incubation experiments. Surface waters (-5 cm) were collected for all canals included in the incubation experiments, and at 5 of the canals we collected water from -10 cm above the canal bottom using gas-tight tubing and a hand pump. Collecting these deeper canal waters for

incubation experiments enabled us to account for any variability in isotopic fractionation of $CH_4$ oxidation with depth in the water column that could impact our estimates of oxidation efficiency. For canal waters collected for incubation experiments, we collected waters as described above but only field acidified samples for the initial incubation time point. Incubations occurred in the dark at room temperature (~25 °C) for 3 days. Duplicate samples for each canal (and depth, if applicable) were acidified every ~24 hours to pH <2 using 1.5 M HCl to stop $CH_4$ oxidation. All incubated waters were analyzed at the Stable Isotope Facility at UC Davis. Dissolved $CH_4$ concentration was calculated as described above.

Dissolved $CH_4$ concentration fell below the limit of quantification within 72 hours, as such incubation results only consider data from the first 2 days. One of the 5 deeper canal waters was omitted as $CH_4$ was not detectable after 24 hours. We calculated potential oxidation rates as the change in $CH_4$ concentration over the total incubation time. We also calculated the fractionation factor of $CH_4$ oxidation, or $\alpha_{ox}$, from the $CH_4$ mixing ratios (in ppm) and $\delta^{13}C\text{-}CH_4$ of the incubated waters using a simplified Rayleigh model[40]:

$$\ln\left(\frac{CH_4}{CH_{4,0}}\right) = \frac{\alpha_{ox}}{1 - \alpha_{ox}} * \ln\left(\frac{1000 + \delta^{13}C - CH_4}{1000 + \delta^{13}C - CH_4, 0}\right) \quad (1)$$

Plotting Eq. (1) with $\ln(1000 + \delta^{13}C\text{-}CH_4)$ on the x-axis and $\ln(CH_4)$ on the y-axis produces a line with a slope of $(\alpha_{ox}/1\text{-}\alpha_{ox})$. As such, we calculated the slope as the difference in $\ln(CH_4)$ between the initial and final time points over the difference in $\ln(1000 + \delta^{13}C\text{-}CH_4)$ over the same time and then solved for $\alpha_{ox}$.

## Canal $CH_4$ emissions

We used a floating chamber to manually collect chamber headspace gasses at 12 of the sampled canals to assess $CH_4$ emissions and emitted $\delta^{13}C\text{-}CH_4$. A 20 cm diameter/2.1 L floating chamber was deployed on the canal water surfaces for 6 minutes in 2023 and 12 minutes in 2024. Chamber deployment time was increased in 2024 to ensure sufficient $CH_4$ accumulation for analysis via Picarro CRDS. The floating chamber was not held in place, but due to low canal water flow (stagnant to ~0.1 m s⁻¹) that chamber did not travel during flux measurement. Three 15 mL gas samples were collected from the chamber headspace over the deployment time via a sampling syringe and injected into a pre-evacuated 12 mL glass Exetainer™ vial (Labco Ltd.). Floating chamber headspace gas samples from 2023 were analyzed at the Stable Isotope Facility at UC Davis and from 2024 at Stanford University, as described above. Methane emissions were calculated as the linear increase in chamber headspace $CH_4$ mixing ratio over the measurement period and converted from ppm $CH_4$ min⁻¹ to mg $CH_4$ m⁻² d⁻¹ using the Ideal Gas Law and the floating chamber dimensions. Fluxes were accepted if the linear increase in $CH_4$ over time met the standards of $R^2 > 0.9$ and $p < 0.05$. Emitted $\delta^{13}C\text{-}CH_4$ was determined via a Keeling plot approach, in which the $\delta^{13}C$ of $CH_4$ emissions is the y-intercept of a linear regression of the inverse mixing ratio of $CH_4$ versus the $\delta^{13}C\text{-}CH_4$ of the corresponding sample[73,74].

We calculated gas transfer velocity (k, m d⁻¹) using data from the subset of canals where paired floating chamber $CH_4$ fluxes and canal water $CH_4$ concentrations were collected using Eq. (2):

$$Flux = k(CH_{4-canal} - CH_{4-eq}) \quad (2)$$

Where $CH_{4\text{-canal}}$ is the concentration of $CH_4$ in canal water, $CH_{4\text{-eq}}$ is the $CH_4$ concentration at equilibrium the atmosphere ($CH_{4\text{-eq}}$), and flux is the rate of $CH_4$ emissions measured using the floating chamber. We used the median k value from the floating chamber deployments to estimate diffusive fluxes across all sampled ($n = 34$) canals. While applying a uniform value introduces uncertainty into the estimates of diffusive fluxes, conditions across the study region are characterized

by high canal water temperature, low canal flow velocity (~0.1 m s⁻¹), and low windspeed. As such, factors that strongly influence $CH_4$ degassing (e.g., solubility and turbulence) should have minimal variation relative to the ~600-fold variation in canal water $CH_4$ concentration across study sites. Values were normalized to $k_{600}$ for literature comparison. See Supplementary Text 1 for further discussion of approaches to estimate k.

## Estimating percent oxidation

We used a simple box model to estimate the percent of $CH_4$ transported from drained peatlands into canals that is oxidized and therefore not emitted to the atmosphere. The model calculated the percent oxidized based on the difference in $\delta^{13}C$ between the source $CH_4$ (e.g., peat porewater) and $CH_4$ after oxidation (e.g., in the canal waters) as well as the isotopic fractionation of $CH_4$ oxidation ($\alpha_{ox}$), which we determined via incubations as described above. Oxidation efficiency ($f_{ox}$) was calculated using a Rayleigh model for closed systems[42,75]:

$$\ln(1 - f_{ox}) = [\ln(\delta_{source} + 1000) - \ln(\delta_{canal} + 1000)]/[\alpha_{ox} - 1] \quad (3)$$

Where $\delta_{source}$ and $\delta_{canal}$ are the $\delta^{13}C\text{-}CH_4$ of peat porewater and drainage canal waters, respectively, and $f_{ox}$ is the fraction of $CH_4$ oxidized. Values of $f_{ox}$ were multiplied by 100 to convert to the percent of $CH_4$ oxidized. The closed system approach represents a lower bound on oxidation, as open system models often result in estimates of the percent oxidized >100%.

The results presented in the main analyses and figures are estimates of the percent oxidized based on mean observed values of $\alpha_{ox}$ (1.022 ± 0.009, from incubations) and $\delta_{source}$ (-85.0 ± 5.9‰, $n = 27$ measurements from 6 porewater profiles, Supplementary Data 2). To characterize the uncertainty of our estimates due to variability in $\alpha_{ox}$ and $\delta_{source}$, we also report how our estimate varies when using ± 1 standard deviation of $\alpha_{ox}$ or $\delta_{source}$ in Eq. (3). Varying $\alpha_{ox}$ or $\delta_{source}$ by ± 1 standard deviation changes our estimate of the mean percent oxidized by ~10%.

To estimate the amount of $CH_4$ emissions attenuated by $CH_4$ oxidation, we back-calculated the concentration of $CH_4$ in canal waters based on the $f_{ox}$ value for each canal:

$$Predicted\ CH_4\ Concentration = \frac{Observed\ CH_4\ Concentration}{1 - f_{ox}} \quad (4)$$

Using this predicted concentration, we calculated predicted diffusive $CH_4$ fluxes as described above. We then subtracted the diffusive $CH_4$ fluxes calculated from observed $CH_4$ concentrations from the predicted $CH_4$ fluxes based on the back-calculated concentrations to estimate the $CH_4$ emissions mitigated by $CH_4$ oxidation.

## Statistical analysis

Statistical analysis and data visualization were performed in R v4.0.3. Data preparation was conducted using the dplyr package[76]. Data and analyses were visualized using the ggplot2[77] and patchwork[78] packages. Statistical analyses were performed using the R Core Team stats package. Dissolved $CH_4$ concentration and estimated diffusive emissions were $\log_{10}$-transformed prior to all statistical analysis to improve normality. If there were replicate measurements taken in a canal, the mean value of replicates was used in statistical analysis and data visualization. Summary statistics were calculated using all observations, including spatial replicates, to report the full range of observations. The level of significance for all analyses was 0.05.

We tested relationships between canal water $CH_4$ concentration and $\delta^{13}C\text{-}CH_4$ and between estimated diffusive $CH_4$ fluxes and the percent of $CH_4$ oxidized using least squares regression. We used Kendall's rank correlation to assess the strength and direction of monotonic relationships between dissolved oxygen or canal depth and

canal water dissolved $CH_4$ concentration, $\delta^{13}C$-$CH_4$, and percent $CH_4$ oxidized. We used a non-parametric correlation for these analyses as relationships between $CH_4$ oxidation and dissolved oxygen are often non-linear due to substrate saturation and potential inhibitory effects of oxygen above the optimal levels for $CH_4$ oxidation in freshwater environments[79]. We used one-way ANOVA to assess the impact of vegetation and land use on canal properties and $CH_4$ cycling.

## Data availability

All data are presented in the manuscript and/or the Supplementary Information. The data used in this study are available at the Zenodo repository under 'Tropical Peatland Drainage Canal Methane Concentrations, Fluxes, and Isotopic Composition' (https://doi.org/10.5281/zenodo.11155160). Source data are provided with this paper.

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

## Acknowledgements

The research was supported by the NSF Earth Sciences Postdoctoral Research Fellowship (Award ID 2305578, CRP) and by the Precourt Institute for Energy (CRP, JCB, AMH). This research was conducted under permit #37/SIP/IV/FR/1/2024 from the Indonesian National Research and Innovation Agency (BRIN). We thank Insen Amri and Anggit Djoko Wibowo

for their assistance with fieldwork and Rob Jackson (Stanford University) for access to the Picarro CRDS used to analyze samples collected in 2024.

## Author contributions

C.R.P., J.C.B., and A.M.H. designed the study. C.R.P., J.C.B., J.S., D.S.P.A.B., E.D., and Y.A. performed the research. C.R.P. analyzed the data and wrote the manuscript. N.N. and G.Z.A. provided field site and laboratory access. J.C.B., J.S., D.S.P.A.B., E.D., Y.A., A.A., A.G., N.N., G.Z.A., and A.M.H. reviewed the manuscript and contributed to manuscript revisions.

## Competing interests

The authors declare no competing interests.
