## [Transparent Peer Review file · Nature Communications]

Fate of methane in canals draining tropical peatlands

Corresponding Author: Dr Clarice Perryman

Version 0:

Reviewer comments:

Reviewer #1

(Remarks to the Author)

This is a well-written, straight to the point manuscript that investigates aerobic methane oxidation in human-made canals draining tropical peatlands. Even though the aim of the study is relatively simplistic, the results are very relevant for a better understanding of methane cycling in these critical systems. The manuscript's objective is clear, methodology is sound, and the data presented answer to the proposed questions. Data displays are of high quality. My major concern is a few overstatements along the text such as L73-74: "Overall, our results indicate CH₄ oxidation is a major control on drainage canal CH₄ emissions in peatlands in Southeast Asia." One cannot claim that peatlands in Southeast Asia in general behave as the 21 canals sampled in a specific region of Indonesia. I suggest editing those sentences to a *potential* important role of methane oxidation in other peatlands across Southeast Asia. Same for L31-32 and parts of the conclusion. I have only a few other comments/questions listed below.

L88: consider adding "on average", 53.5%...

L101-103: this claim depends on incubation temperatures. What was the incubation temperature in this study? In situ temperature?

L128: remove "in canals". Not needed as you start with "canal water..."

L157-159: please add how much the oxidation mitigation represents in terms percentage.

L173-174: yes, but you have a very narrow range of dissolved oxygen! Your whole range falls into hypoxic.

L191-192: Why is that? Is it possible that methane oxidation consumes the O₂ produced by plants in a cryptic cycle (sensors don't capture the availability of O₂)? Please discuss.

L250-251: what value of fractionation factor was used to calculate percent CH₄ oxidized from the δ¹³C-CH₄?

L294: how do the estimates of CH₄ flux based on wind data of a meteorological station compares to the floating chamber measurements you've done? Are there significant differences in the calculated oxidation mitigation if you use one or the other method of flux estimation?

L321-322: Can you report the R² of these relationships? Methane oxidation usually follows a 1st order reaction, meaning that the natural logarithm of CH₄ concentration shows a linear relationship with time and the slope of that relationship is the rate constant of oxidation.

Reviewer #2

(Remarks to the Author)

The manuscript by Perryman and colleagues addresses methane emissions from peatland ditches in Southeast Asia. They measured methane concentrations and isotopic composition in 20 ditches/canals (once) and conducted incubation experiments for a subset of these ditches. They find that methane oxidation within the water column consumes ~75% of methane, and thus net emissions from the ditches are significantly decreased by this "biofilter".

Overall, I enjoyed reading the paper. The writing is good, the figures nicely drawn, and the results support the conclusions. The research topic is important: we know that ditches emit large amounts of CH₄ but we lack data from tropical peatlands. Additionally, most studies only consider net emission from the water surface, so process studies such as this add novel information.

That said, some parts of the manuscript slightly let it down. I was disappointed not to find raw data linked. Apparently this will be uploaded upon acceptance but it would have been useful to be able to evaluate it during my review.

Another issue is the relatively small sample number. The authors measured 20 canals, used 13 for incubation experiments, 5 for a second depth of incubation experiments (was this data reported in the MS?), then 8 for floating chambers. So the experimental design is slightly messy. That's fine – fieldwork often runs that way. But using this small sample size (measured just once) the authors occasionally make quite sweeping statements: “our finding [...] is likely robust across peatland drainage canals in Southeast Asia.” Considering industrial oil palm ditches are not well represented in the data, which focuses mostly on smallholder land, and the lack of temporal replication, this is quite a claim.

Also considering the sampling design, it seems that (assuming I interpret correctly) there is some pseudoreplication of sample points in some of the scatter plots, where duplicate measurements are presented separately. This is easily fixed (assuming my interpretation is correct).

Another minor issue relates to IPCC accounting. Ditch CH₄ emissions are accounted for in the 2019 IPCC Refinement and (more relevant to this study) in the 2013 Wetlands Supplement. The Wetlands Supplement (Table 2.4*) (and associated paper <https://doi.org/10.1007/s00027-015-0447-y>) highlighted a lack of data from tropical peat ditches but did give an EF. It would be interesting to know how your emissions compare to the IPCC EF (they're lower I think, if my conversions are correct), and also worth highlighting that ditch emissions are anthropogenic and should be accounted for in inventories. *<https://www.ipcc.ch/publication/2013-supplement-to-the-2006-ipcc-guidelines-for-national-greenhouse-gas-inventories-wetlands/>

Note that I do not have experience of running isotope/oxidation experiments. I assume methods and analysis here are fine, but cannot comment with any authority myself.

Following revision, I think the manuscript would be acceptable for publication in Nature Communications.

Mike Peacock

Line comments follow

The abstract is concisely written. However, you write:

“We find that CH₄ oxidation mitigates potential canal CH₄ emissions by $75.5 \pm 12.8\%$, reducing CH₄ emissions by 24.3 ± 32.3 mg CH₄ m⁻² d⁻¹”

Many people reading will want to know the headline figure of mean CH₄ emission without having to do any maths themselves. So you could rephrase to:

“We find that CH₄ oxidation mitigates potential canal emissions by $75.5 \pm 12.8\%$, reducing mean emissions from XX to YY mg CH₄ m⁻² d⁻¹”

Additionally, if word limits allow I think you could add a few words into the abstract to say something about how many sites/canals you sampled, and if it was temporally replicated or just a synoptic snapshot.

Also, there's a lot of “CH₄ emissions” in this abstract. Once you've established that it's CH₄ we're talking about then it's fine (and will save you a few words) to just write “emissions”.

L35. “Wetlands and freshwaters contribute ~30-55% of global CH₄ emissions, largely from the tropics”

Is this perhaps a bit of a stretch? Tropical wetlands are well implicated in rising atmospheric CH₄ emissions, but northern waterbodies are also high emitters (as discussed by Rocher-Ros et al, who you cite here). And Rosentreter et al, in their global synthesis of aquatic CH₄ say:

“Despite the global coverage of our data, we did not detect clear latitudinal trends of methane emissions from aquatic ecosystems, except for the emissions from coastal wetlands peaking at 30° N.”

So I would suggest toning down this sentence.

L49. “Given the increased importance of aquatic carbon fluxes in drained tropical peatlands...”

To me, this sentence could be clearer. Is this increased importance compared to the past (due to global change), or compared to undrained tropical peatlands, or to peatlands in other climate zones?

L62. “Constraining the importance of CH₄ oxidation in drainage canals in tropical peatlands is a key step to improving CH₄ budgets of these ecosystems.”

Is it? Isn't simply measuring emissions the key step to improving budgets? What's oxidised within the water column is irrelevant to the budget (but of course is interesting nonetheless).

L68. I read the intro first, then methods, and was frustrated to find a lack of info on the canal reaches measured. But then eventually in the results Table S1 is mentioned. This should also be mentioned around L68 (study sites) and L275 (methods) so the reader knows this info is available.

L68. Here you say 21 canal reaches, but Table S1 shows 20 (presumably because 34 was measured twice). Seems like 20

reaches is the correct one. This also applies to the “field sampling” section of methods.

L88. “Across incubated waters, $53.5 \pm 26.0\%$ of the initial CH₄ was consumed over the incubation period (17.6%-99.7%, Fig. 2A).”

This is interesting, and the following lines give info on general conc changes, but it would be nice to see a multi-panel figure in the SI showing raw conc changes for each incubation.

L96. “averaged”. Also L140, 178, L229. Be clear if these are means (presumably) or medians.

L137. “It is unlikely that CH₄ concentration in canal waters is dictated only by the amount of CH₄ originally transported into canals from the surrounding landscape.”

Agreed, but in-situ production could also be relevant. Three decades (!) ago Roulet and Moore considered the potential importance of lateral transport vs in-situ production <https://cdns.ciencepub.com/doi/abs/10.1139/x95-055>

L161. There are 35 data points for 21 canal reaches (and presumably the same for Fig 4a too). It isn't obvious where this 35 comes from. The “incubations” section mentions sample duplicates but I don't think the “canal CH₄” section does. Also, if you're plotting sample duplicates as independent data points isn't this essentially pseudoreplication which will artificially inflate your sample size and therefore statistical power? It seems more honest to calculate these correlations on means of the duplicates.

L189. “As such, shallow canal water depths likely support more CH₄ oxidation.”

This is interesting. Other studies, including of ditches (e.g. <https://link.springer.com/article/10.1007/s13157-011-0170-y>) have found that CH₄ emissions are lower in deeper ditches, because there is more scope for oxidation within the water column. There's also ebullition to consider, which is likely to be higher in shallow canals (because sediments will become warmer in these systems).

L208. “our finding that the majority of CH₄ transported into drainage canals is oxidized rather than emitted is likely robust across peatland drainage canals in Southeast Asia.”

I acknowledge that it's always nice to claim that research findings are widely applicable but this is, to me, overreaching. You measured 20 canals, once, in two study regions. I would suggest toning down this sentence.

L218. Somewhere in the paper, and here seems a good place for it, it would be good to give some mean fluxes from other tropical ditch studies for comparison.

L276. This says 100cm was your max water depth, Table S1 says 71cm.

L277. Please add canal widths to Table S1.

L278. “and canals situated on peatlands under a variety of land uses (smallholder mixed agriculture, smallholder oil palm, and industrial oil palm) to capture the heterogeneity of drainage canals in the region”

Although you only have three in industrial oil palm. Perhaps worth emphasising that your study is biased towards smallholder systems?

L298. If you're calculating k solely using windspeed I guess that means these canals aren't flowing (i.e. there is zero turbulence). If so, best to explicitly say so in the text. But I admit to getting a little hesitant when using wind speed to calculate emissions in these small waterbodies. These relationships between wind speed and k have been tested in lakes, but can you be sure they are appropriate for small, sheltered ditches (e.g. Fig. 1d)? I would suggest:

1. Including your k values somewhere so the reader can check them. How do they compare to k values from other inland waters and from other ditches (or small waterbodies, e.g. ponds)?
2. Can you calculate some k values from your floating chamber deployments? Do they compare to your windspeed-derived k values well?

There is some data in Fig. S4 where it seems like chambers are giving higher fluxes compared to k values. Is this due to problems with the k method, or are chambers capturing bubble events too? Do you see any evidence of ebullition in these ditches?

L307. “We collected canal waters at a subset (n = 13) of the drainage canals for incubation experiments”
Which canals? Can you add this into to Table S1.

L309. How did you collect the deeper water samples? Please state.

L315. “Duplicate samples for each canal (and depth, if applicable) were acidified every ~24 hours to pH < 2 using 1.5M HCl to stop CH₄ oxidation.”

So you had two replicates for each measurement and then (presumably) took the mean of both? This is good, but it would be nice to see the reps data; how consistent are they to one another? Considering your small sample size this info would be useful.

L338. What depth pore water? Please state.

Reviewer #3

(Remarks to the Author)
Comments attached

Version 1:

Reviewer comments:

Reviewer #1

(Remarks to the Author)
I am satisfied with the responses and changes made to the manuscript.

Reviewer #2

(Remarks to the Author)
Overall, the authors have thoroughly considered my original comments and revised their manuscript accordingly. I certainly didn't require, or expect, additional data to appear, but the new data from 13 canal reaches in another area further strengthen the small dataset and are a welcome addition. I have two small comments on this new draft. Otherwise, I find the manuscript acceptable for publication and look forward to seeing the published version.

Original comment:

L315. "Duplicate samples for each canal (and depth, if applicable) were acidified every ~24 hours to pH <2 using 1.5M HCl to stop CH₄ oxidation."

So you had two replicates for each measurement and then (presumably) took the mean of both? This is good, but it would be nice to see the reps data; how consistent are they to one another? Considering your small sample size this info would be useful.

Author response:

We have added a supplementary table (Table S2) that has the initial and final CH₄ concentrations and $\delta^{13}\text{C-CH}_4$ (mean \pm standard error) and the incubation time for all incubated waters.

New comment:

My original comment asked for the replicate sample data to be included – that is still hidden in Table S2 by the use of means (although the SEM values give hints). It isn't in the online data either: the file Canal_Water_Incubations_Perryman has the replicate measurements of dissolved CH₄ and d¹³CH₄ (or are these the replicate *changes* in these parameters?) but this isn't sufficient. To be clear, I would like to see the raw, replicate data, set out as in Table S2 (CH₄ T₀, CH₄ T_{final}, ¹³C T₀, ¹³C T_{final}) whereby each individual, replicate incubation has its own line (i.e. not averaged together) – unless these data are already hiding somewhere in the SI but if so I don't see it. It's potentially important/interesting for the reader to see how consistent your reps are.

One minor comment

L344. "Open undeveloped land" is slightly vague because to a casual reader it hides human action. Perhaps change to "deforested undeveloped land" (or similar)?

Mike Peacock

Reviewer #3

(Remarks to the Author)
Comments on the reviewed manuscript NCOMMS-24-34376 "Fate of methane in canals draining tropical peatlands" by Perryman et al.

Perryman et al. have done an amazing job addressing my concerns in the revised manuscript. All the points raised were carefully considered in the revised text, including new relevant information and clarifications. I don't have any further considerations about the manuscript and I believe this is a valuable contribution to the field. Similarly to the manuscript, I would like to thank the authors for this well-structured and nice to read response letter.

We appreciate the thorough and constructive feedback on our manuscript from all 3 reviewers and feel
our manuscript has been strengthened through the revisions made in response to their suggestions. We
provide a detailed response to all comments below. Here, we summarize the major revisions to our
manuscript. **These revisions improved the manuscript, but did not change the key finding that CH₄**
**oxidation significantly attenuates CH₄ emissions from canals draining peatlands in the study**
**region.** We feel that the revisions increased the scientific rigor of the work and we are grateful for the time
and care all 3 reviewers invested in helping us improve the manuscript.

Firstly, the revised manuscript includes new data from 13 additional sampling sites. Adding these new
data addresses the shared concern amongst all 3 reviewers regarding the sample size and
representativeness of the original dataset. The sample size for the revised manuscript is 34 canal
reaches, compared to 21 canal reaches in the original manuscript. These new data also improve the
spatial coverage and land use representation in our study, as they are from canals in another drained
peatland area (~50-100 km away from the sites presented in the original manuscript) and increase the
representation of industrial oil palm, smallholder plantations, and open undeveloped land. The data
collected at these new sites are consistent with observations presented in the original manuscript. These
include CH₄ concentrations, $\delta^{13}\text{C-CH}_4$, basic water chemistry, and canal dimensions for all sites, and
floating chamber CH₄ emissions for a subset of sites. As the expanded dataset shows the same trends in
CH₄ concentrations, isotopes, and fluxes as the original dataset, we feel that the inclusion of these new
data further support that CH₄ oxidation has a significant influence on emissions from canals draining
peatlands in Southeast Asia.

In response to reviewer feedback, we have adapted language throughout the text to avoid overstating the
implications of our study. While the expansion of the dataset strengthened our results, we acknowledge
that the data only includes sites in West Kalimantan. However, given the similarity in the physical and
chemical properties of canals draining peatlands across Southeast Asia (see response to Reviewer 2 on
line 259-270 of this document), we feel the study sites in our work are representative of the broader
region. We have summarized results of canal CH₄ concentrations and emissions from canals across the
major peatland regions in Southeast Asia in Tables S7-8, showing that our sample sites are similar to
canals across Borneo, Peninsular Malaysia, and Sumatra draining peat soils under varying land uses.
The lack of other isotopic datasets to which we could compare our results highlights the research gap our
study aims to fill by providing isotopically-enabled insights to CH₄ processing in canal waters.

Secondly, in response to comments from all 3 reviewers we have revised our approach to estimating
diffusive CH₄ emissions and have added more methodological detail in the main and supplementary texts.
In the original manuscript, we modeled gas transfer velocity from wind speed. In the revised manuscript
we determined gas transfer velocity from floating chamber deployments at a subset of study sites. The
revised approach provides a more site- and CH₄-specific estimate of gas transfer velocity. This revision
increased our estimate of diffusive CH₄ emissions, and the revised diffusive fluxes are in better
agreement with observations from floating chamber deployments at our study sites and prior work in other
regions of Southeast Asia. Supplementary Text 1 of the revised manuscript discusses how approaches to
estimating diffusive fluxes impact our results, and we include a new supplementary table (Table S8)
comparing canal CH₄ emissions across tropical peatlands.

Thirdly, the revised manuscript includes more discussion of other potential factors that could influence our
isotopic results, and therefore our estimates of the efficiency of methane oxidation in drainage canal
waters. This includes: potential environmental correlates of isotopic fractionation (L106-113), methane
production in canal sediments and canal waters (L149-168, L217-228), seasonality (L247-259), and

variability in methane production pathways and source methane isotopic composition across the
 landscape (L151-159, L217-228).

 The table below summarizes the key revisions made to the manuscript. Further discussion of these
 changes and other reviewer concerns can be found below. We have uploaded the data used to generate
 the figures and results presented in the manuscript to a Zenodo repository:
 <https://doi.org/10.5281/zenodo.11155160>

	Original Manuscript	Revised Manuscript	Change made
Number of canal reaches	21	34	Added new data additional sites
Land use representation	Smallholder mixed agriculture, industrial oil palm, smallholder plantation	Smallholder mixed agriculture, industrial oil palm, smallholder plantation, open undeveloped, degraded forest	Added new data additional sites and land use types
Estimated percent oxidized	75.5 ± 12.8% (n = 35 observations from 21 canal reaches)	76.4 ± 12.0% (n = 48 observations from 34 canal reaches)	Added new data from additional sites
Estimated diffusive fluxes	16.1 ± 33.2 CH ₄ m ⁻² d ⁻¹ (n = 35 observations from 21 canal reaches)	72.2 ± 151.2 CH ₄ m ⁻² d ⁻¹ (n = 48 observations from 34 canal reaches)	Revised approach to estimating gas transfer velocity and added new data from additional sites
Floating chamber fluxes	98.9 ± 153.7 CH ₄ m ⁻² d ⁻¹ (n = 12 observations from 7 canal reaches)	94.9 ± 142.3 CH ₄ m ⁻² d ⁻¹ (n = 18 observations from 12 canal reaches)	Added new data from additional sites

Reviewer #1 (Remarks to the Author):

We thank Reviewer 1 for their feedback on our manuscript. To address their main concerns, we have
 tempered some statements in the manuscript to avoid overstating the implications of our study. We feel
 the new data added during revision strengthen our finding that CH₄ oxidation is a major influence on canal
 CH₄ emissions by showing consistent results across peatland areas ~50-100 km apart, but we
 acknowledge the potential limitations of the findings given the spatial extent of our sampling. We also
 modified our approach to calculating oxidation rates from the laboratory incubations in response to their
 question about the incubation data. This resulted in a negligible change to the results. Please see below
 for detailed responses to all comments raised by Reviewer 1.

This is a well-written, straight to the point manuscript that investigates aerobic methane oxidation in
 human-made canals draining tropical peatlands. Even though the aim of the study is relatively simplistic,
 the results are very relevant for a better understanding of methane cycling in these critical systems. The
 manuscript's objective is clear, methodology is sound, and the data presented answer to the proposed
 questions. Data displays are of high quality. My major concern is a few overstatements along the text
 such as L73-74: "Overall, our results indicate CH₄ oxidation is a major control on drainage canal CH₄

emissions in peatlands in Southeast Asia.” One cannot claim that peatlands in Southeast Asia in general
behave as the 21 canals sampled in a specific region of Indonesia. I suggest editing those sentences to a
*potential* important role of methane oxidation in other peatlands across Southeast Asia. Same for L31-
32 and parts of the conclusion.

We have revised summary statements in the manuscript to avoid overstatements, while still highlighting
the implications of our findings for CH₄ emissions from canals and the tropical peatlands they drain. For
example:

L31-33: *“As canals drain over 65% of peatlands in Southeast Asia, our results suggest that CH₄ oxidation
significantly influences landscape-scale CH₄ emissions from these ecosystems.”*

L73-74: *“Overall, our results suggest that CH₄ oxidation substantially attenuates CH₄ emissions from
canals”*

L317-323: *“In summary, we demonstrate that CH₄ oxidation can substantially attenuate CH₄ emissions
from canals draining peatlands in Southeast Asia. We estimate that CH₄ oxidation mitigates >50% of
potential CH₄ emissions from canals across West Kalimantan, Indonesia. As landscape-scale
measurements of CH₄ exchange in drained tropical peatlands indicate that canal networks contribute
disproportionately to emissions from these ecosystems¹⁸, our results suggest that CH₄ oxidation
influences emissions not only from drainage canals but from degraded peatlands in Southeast Asia as a
whole.”*

I have only a few other comments/questions listed below.

L88: consider adding “on average”, 53.5%...

We have revised this sentence and corrected a typo (mean = 53.8%, not 53.5%, L90):

*“On average, 53.8 ± 25.6% of the initial CH₄ was consumed over the incubation period (17.6%-99.7%)
and δ¹³C-CH₄ increased by 19.8 ± 17.7‰ (2.1-67.8‰, Fig. 2A).”*

L101-103: this claim depends on incubation temperatures. What was the incubation temperature in this
study? In situ temperature?

These lines of the manuscript were omitted during revision.

L128: remove “in canals”. Not needed as you start with “canal water...”

We integrated this suggestion into the revised text.

L157-159: please add how much the oxidation mitigation represents in terms percentage.

This estimate of CH₄ emissions mitigated by oxidation is based off of the estimates of percent oxidized
presented in the first paragraph of this section of the paper, therefore this information is given earlier in
this section. L124-125:

*“...we estimated that CH₄ oxidation consumes 76.4 ± 12.0% of CH₄ transported into canals (range: 47.3-
91.3%).”*

L173-174: yes, but you have a very narrow range of dissolved oxygen! Your whole range falls into
hypoxic.

We have added additional context here for our interpretation that CH₄ oxidation in canals is mediated by
aerobic methanotrophs (L196-206):

*“We found that the percent of CH₄ oxidized increased and dissolved CH₄ concentration decreased with
the concentration of dissolved oxygen at the canal water surface (0-10 cm; p < 0.05, Fig. 4A, Table S4).
The relationship between dissolved oxygen and CH₄ oxidation is consistent with oxidation mediated by
aerobic methanotrophic bacteria, as has been observed in other stream and river networks^{31,45}. While all
canals had low dissolved oxygen (0.2 to 2.3 mg L⁻¹), methanotrophic bacteria of the order
Methylococcales have been shown to have the genetic potential for survival and methanotrophic activity
in low oxygen environments⁴⁹. Abundant Methylococcales have been identified in hypoxic tropical
freshwaters where paired measurements of dissolved CH₄ concentration and δ¹³C-CH₄ indicate ongoing
CH₄ oxidation^{33,34}. Our results further support the idea that aerobic CH₄ oxidation occurs in tropical
freshwaters with low dissolved oxygen.”*

L191-192: Why is that? Is it possible that methane oxidation consumes the O₂ produced by plants in a
cryptic cycle (sensors don't capture the availability of O₂)? Please discuss.

We have added the following text to L208-214:

*“Vegetation may enhance CH₄ oxidation via radial oxygen loss from roots^{50,51} or via oxidation by epiphytic
methanotrophs in submersed plants⁵². Although we did not observe a significant difference in dissolved
oxygen based on the presence of aquatic vegetation (p > 0.05, Table S5), oxygen delivered to the water
column by aquatic vegetation is likely rapidly consumed by methanotrophs or by competing aerobic
heterotrophs as deposition of more labile organic carbon by aquatic vegetation could stimulate
heterotrophic respiration in canal waters²⁸.”*

L250-251: what value of fractionation factor was used to calculate percent CH₄ oxidized from the d13C-
CH₄?

We have added this information to the figure caption for Figure 5:

*“The percent of CH₄ oxidized in drainage canal waters (estimated from dissolved δ¹³C-CH₄ using α_{ox} =
1.022) versus the δ¹³C of CH₄ emitted from the corresponding canal.”*

L294: how do the estimates of CH₄ flux based on wind data of a meteorological station compares to the
floating chamber measurements you've done? Are there significant differences in the calculated oxidation
mitigation if you use one or the other method of flux estimation?

We revised our approach to estimating fluxes (using gas exchange velocity from floating chambers rather
than wind speed) in response to feedback from Reviewer 2 and 3. Please see L417-431 for an
explanation of the revised approach:

*“We calculated gas transfer velocity (k, m d⁻¹) using data from the subset of canals where paired floating
chamber CH₄ fluxes and canal water CH₄ concentrations were collected using Eqn. (2):*

$$162 \text{Flux} = k(\text{CH}_{4\text{-canal}} - \text{CH}_{4\text{-eq}}) \quad (\text{Eqn. 2})$$

*Where CH_{4-can} is the concentration of CH₄ in canal water, CH_{4-eq} is the CH₄ concentration at equilibrium
the atmosphere (CH_{4-eq}), and flux is the rate of CH₄ emissions measured using the floating chamber. We
used the median k value from floating chamber deployments to estimate diffusive fluxes across all
sampled (n = 34) canals. While applying a uniform value introduces uncertainty into the estimates of
diffusive fluxes, conditions across the study region are characterized by high canal water temperature,*

*low canal flow velocity (~0.1 m s⁻¹), and low windspeed. As such, factors that strongly influence CH₄*
*degassing (e.g., solubility and turbulence) should have minimal variation relative to the ~600-fold variation*
*in canal water CH₄ concentration across study sites. Values were normalized to k₆₀₀ for literature*
*comparison. See Supplementary Section 1 for further discussion of approaches to estimate k.”*

Responses to the other 2 reviewers below provide further discussion of this change. The estimated fluxes
in the revised manuscript are higher, thus so are the estimates of emissions mitigated by CH₄ oxidation.

L321-322: Can you report the R² of these relationships? Methane oxidation usually follows a 1st order
reaction, meaning that the natural logarithm of CH₄ concentration shows a linear relationship with time
and the slope of that relationship is the rate constant of oxidation.

We have streamlined our approach to calculating potential oxidation rates in the revised manuscript. To
mitigate issues with linearity, we revised our calculations to estimate potential oxidation rates as the
difference in the initial and final CH₄ concentrations (mean of 2 replicates for each time point) divided by
the incubation time. For consistency, we also revised our calculation of α_{ox} to use just the initial and final
time points. Denfeld et al. (2016; *JGR Biogeosciences*) previously used initial and final time points to
calculate α_{ox} in lakes using similar calculations. We have clarified this change in our approach in L388-
397:

*“We calculated potential oxidation rates as the change in CH₄ concentration over the total incubation*
*time. We also calculated the fractionation factor of CH₄ oxidation, or α_{ox}, from the CH₄ mixing ratios (in*
*ppm) and δ¹³C-CH₄ of the incubated waters using a simplified Rayleigh model³⁹:*

$$\ln(1 - f_{ox}) = [\ln(\delta_{source} + 1000) - \ln(\delta_{canal} + 1000)] / [\alpha_{ox} - 1] \quad (\text{Eqn. 3})$$

*Plotting Eqn. 1 with ln(1000 + δ¹³C-CH₄) on the x-axis and ln(CH₄) on the y-axis produces a line with a*
*slope of (α_{ox}/1-α_{ox}). As such, we calculated the slope as the difference in ln(CH₄) between the initial and*
*final time points over the difference in ln(1000 + δ¹³C-CH₄) over the same time, and then solved for α_{ox}.”*

The change in our approach resulted in very minor changes to the incubation results. Critically, as the
main findings on our paper are highly dependent on α_{ox}, the change to the mean α_{ox} value was negligible,
now reporting 1.022 ± 0.009 vs. 1.022 ± 0.008.

	Oxidation Rate - umol CH ₄ L ⁻¹ d ⁻¹	α _{ox} - Mean (Range)
Original Draft	0.3 to 6.6	1.022 ± 0.008 (1.002 to 1.035)
Revised Manuscript	0.3 to 5.6	1.022 ± 0.009 (1.002 to 1.039)

Reviewer #2 (Remarks to the Author):

We appreciate Dr. Peacock’s thorough and thoughtful comments on our manuscript. We have made
several revisions in response to the concerns raised by Dr. Peacock:

- 1. We have increased the sample size for estimating the fraction oxidized from 21 canal reaches to
34 canal reaches with the addition of 13 new sample sites. Adding these data also improved the
representation of canals in industrial oil palm plantations, among other land uses, in our dataset
and expanded the spatial coverage of our study.

- 2. We have revised our statistical approach and conducted statistical analyses using mean values in
cases where replicate measurements were collected. All revised data visualizations were also
produced from mean values.
- 3. We revised our approach to estimating diffusive CH₄ fluxes. In the revised manuscript we report
fluxes calculated using gas transfer velocity determined through the floating chamber
deployments. We added additional discussion of our approach to estimating fluxes in the main
text (L417-431) and in Supplementary Text 1. Please see further discussion of these revisions in
response to comments from both Dr. Peacock's and Reviewer 3 comments below.
- 4. The revised manuscript includes discussion text (L269-277) and a supplementary table (Table
S8) comparing the canal emissions we observed to past work in other regions of Southeast Asia.
- 5. The data presented in the paper are now available at: <https://doi.org/10.5281/zenodo.11155160>

The incorporation of the new data and revised statistics/calculations did not change our key finding that
CH₄ oxidation substantially attenuates emissions from canals draining peatlands in Southeast Asia, but
we do feel that the revisions bolstered our efforts to rigorously assess these results. Please find our
detailed responses to all of Dr. Peacock's comments below.

The manuscript by Perryman and colleagues addresses methane emissions from peatland ditches in
Southeast Asia. They measured methane concentrations and isotopic composition in 20 ditches/canals
(once) and conducted incubation experiments for a subset of these ditches. They find that methane
oxidation within the water column consumes ~75% of methane, and thus net emissions from the ditches
are significantly decreased by this "biofilter".

Overall, I enjoyed reading the paper. The writing is good, the figures nicely drawn, and the results support
the conclusions. The research topic is important: we know that ditches emit large amounts of CH₄ but we
lack data from tropical peatlands. Additionally, most studies only consider net emission from the water
surface, so process studies such as this add novel information.

That said, some parts of the manuscript slightly let it down. I was disappointed not to find raw data linked.
Apparently this will be uploaded upon acceptance but it would have been useful to be able to evaluate it
during my review.

The data are available at: <https://doi.org/10.5281/zenodo.11155160>

Another issue is the relatively small sample number. The authors measured 20 canals, used 13 for
incubation experiments, 5 for a second depth of incubation experiments (was this data reported in the
MS?), then 8 for floating chambers. So the experimental design is slightly messy. That's fine – fieldwork
often runs that way. But using this small sample size (measured just once) the authors occasionally make
quite sweeping statements: "our finding [...] is likely robust across peatland drainage canals in Southeast
Asia." Considering industrial oil palm ditches are not well represented in the data, which focuses mostly
on smallholder land, and the lack of temporal replication, this is quite a claim.

The points raised by all reviewers about tempering the language used in some statements is well taken.
To address the specific points about sample size here, we have added new data to the revised
manuscript. The revised manuscript includes data from 13 canal reaches in a second peatland area ~50-
100 km north of the 21 canals included in the original manuscript. These data include additional
measurements of CH₄ fluxes and ¹³C from canals from oil palm plantations, as well as canals from "open
undeveloped" (i.e., deforested but no active land use) areas. These samples were collected in April 2024
vs. the data presented in the original manuscript which came from fieldwork conducted in May 2023.

We acknowledge the limitations of even this expanded dataset, but feel the inclusion of these data
support our results that CH₄ oxidation is an important control on canal CH₄ emissions. Table S7 and S8
report results for dissolved CH₄ and oxygen concentrations and CH₄ emissions from canals in tropical
peatlands across Indonesia, Malaysia, and Brunei. Our study canals are within the range observed
across the region for all parameters, including canals from land uses not represented in our work and
sampled during more pronounced wet or dry periods than our study. As we identified dissolved oxygen as
a significant correlate of CH₄ oxidation, we feel there is merit in our assessment that CH₄ oxidation is
likely prevalent in canals across the wider region. Furthermore, canals across Southeast Asia are similar
in their physical (depth, width) and chemical (low pH, high DOC concentration and aromaticity, low
dissolved oxygen, etc.) characteristics (Bowen et al., 2024, *Nature Geoscience*; Extended Data Table 2,
Extended Data Figure 4), further supporting that the conditions under which CH₄ oxidation occurs in
canals are broadly consistent across the study region.

Also considering the sampling design, it seems that (assuming I interpret correctly) there is some
pseudoreplication of sample points in some of the scatter plots, where duplicate measurements are
presented separately. This is easily fixed (assuming my interpretation is correct).

We have revised our statistical analyses. The statistics and data visualization presented in the revised
manuscript are based off of the mean values for each canal to avoid pseudoreplication. Summary
statistics (means, ranges, etc.) reported in the revised paper are based on all observations (n = 48
observations from 34 canal reaches for canal water CH₄ measurements) including spatial replicates to
report the full range of observations. We include data files of both the mean values and all spatial
replicates in the files uploaded to our data repository.

Another minor issue relates to IPCC accounting. Ditch CH₄ emissions are accounted for in the 2019
IPCC Refinement and (more relevant to this study) in the 2013 Wetlands Supplement. The Wetlands
Supplement (Table 2.4*) (and associated paper <https://doi.org/10.1007/s00027-015-0447-y>) highlighted a
lack of data from tropical peat ditches but did give an EF. It would be interesting to know how your
emissions compare to the IPCC EF (they're lower I think, if my conversions are correct), and also worth
highlighting that ditch emissions are anthropogenic and should be accounted for in inventories.

*<https://www.ipcc.ch/publication/2013-supplement-to-the-2006-ipcc-guidelines-for-national-greenhouse-gas-inventories-wetlands/>

We have added a comparison to the IPCC EF in the discussion, and noted that canal emissions are
anthropogenic (L269-277):

*“Our observations of canal CH₄ emissions estimated from dissolved CH₄ concentration (72.2 ± 151.2 mg
CH₄ m⁻² d⁻¹) and collected using floating chambers (94.9 ± 142.3 mg CH₄ m⁻² d⁻¹) are within range of past
observations from Indonesia^{25,26,60} and Malaysia²⁴ where mean emissions range from 2.8 to 1073 mg CH₄
m⁻² d⁻¹ (Table S8). The IPCC CH₄ Emissions Factor for canals in tropical peatlands of 618.9 mg CH₄ m⁻²
d⁻¹ (2259 kg CH₄ ha⁻¹ y⁻¹) was based on the only reported data²⁵ at the time of the 2013 Wetlands
Supplement⁶¹ This emission factor now represents the high end of field estimates to date among a still
small number of existing studies and should be reconsidered to more accurately inventory the
anthropogenic (e.g., from land use change) component⁶² of CH₄ emissions from degraded tropical
peatlands.”*

Note that I do not have experience of running isotope/oxidation experiments. I assume methods and
analysis here are fine, but cannot comment with any authority myself.

Following revision, I think the manuscript would be acceptable for publication in Nature Communications.

Mike Peacock

Line comments follow

The abstract is concisely written. However, you write:

“We find that CH₄ oxidation mitigates potential canal CH₄ emissions by 75.5 ± 12.8%, reducing CH₄
emissions by 24.3 ± 32.3 mg CH₄ m⁻² d⁻¹”

Many people reading will want to know the headline figure of mean CH₄ emission without having to do
any maths themselves. So you could rephrase to:

“We find that CH₄ oxidation mitigates potential canal emissions by 75.5 ± 12.8%, reducing mean
emissions from XX to YY mg CH₄ m⁻² d⁻¹”

Additionally, if word limits allow I think you could add a few words into the abstract to say something
about how many sites/canals you sampled, and if it was temporally replicated or just a synoptic snapshot.

Also, there’s a lot of “CH₄ emissions” in this abstract. Once you’ve established that it’s CH₄ we’re talking
about then it’s fine (and will save you a few words) to just write “emissions”.

*We have streamlined the abstract in order to fit in the information that this was a synoptic survey of 34
canals (with the addition of new study sites). The abstract now reads:*

*“Tropical wetlands and freshwaters are major contributors to the growing atmospheric methane (CH₄)
burden. Extensive peatland drainage has lowered CH₄ emissions from peat soils in Southeast Asia, but
the canals draining these peatlands may be hotspots of CH₄ emissions. Alternatively, CH₄ consumption
(oxidation) by methanotrophic microorganisms may attenuate emissions. We used laboratory experiments
and a synoptic survey of the isotopic composition of CH₄ in 34 canals across West Kalimantan, Indonesia
to quantify the proportion of CH₄ that is consumed and therefore not emitted to the atmosphere. We find
that CH₄ oxidation mitigates 76.4 ± 12.0% of potential canal emissions, reducing emissions by ~70 mg
CH₄ m⁻² d⁻¹. Methane consumption also significantly impacts the stable isotopic fingerprint of canal CH₄
emissions. As canals drain over 65% of peatlands in Southeast Asia, our results suggest that CH₄
oxidation significantly influences landscape-scale CH₄ emissions from these ecosystems.”*

L35. “Wetlands and freshwaters contribute ~30-55% of global CH₄ emissions, largely from the
tropics”

Is this perhaps a bit of a stretch? Tropical wetlands are well implicated in rising atmospheric CH₄
emissions, but northern waterbodies are also high emitters (as discussed by Rocher-Ros et al, who you
cite here). And Rosentreter et al, in their global synthesis of aquatic CH₄ say:

“Despite the global coverage of our data, we did not detect clear latitudinal trends of methane emissions
from aquatic ecosystems, except for the emissions from coastal wetlands peaking at 30° N.”

So I would suggest toning down this sentence.

*We have revised this sentence to say (L35-36):*

*“Wetlands and freshwaters contribute ~30-55% of global CH₄ emissions¹, with significant emissions from
tropical ecosystems²⁻⁴.”*

L49. “Given the increased importance of aquatic carbon fluxes in drained tropical peatlands...”

To me, this sentence could be clearer. Is this increased importance compared to the past (due to global
change), or compared to undrained tropical peatlands, or to peatlands in other climate zones?

*We have revised this sentence to say (L50-51):*

*“Given that drainage increases the importance of aquatic carbon fluxes from tropical peatlands...”*

L62. “Constraining the importance of CH₄ oxidation in drainage canals in tropical peatlands is a key step
to improving CH₄ budgets of these ecosystems.”

Is it? Isn't simply measuring emissions the key step to improving budgets? What's oxidised within the
water column is irrelevant to the budget (but of course is interesting nonetheless).

We have revised this sentence to say (L62-64):

*“Constraining the importance of CH₄ oxidation in canals draining tropical peatlands is a key step to
improving our understanding of the processes controlling CH₄ emissions from these ecosystems...”*

L68. I read the intro first, then methods, and was frustrated to find a lack of info on the canal reaches
measured. But then eventually in the results Table S1 is mentioned. This should also be mentioned
around L68 (study sites) and L275 (methods) so the reader knows this info is available.

We added references to Table S1 in the sections noted here.

L68. Here you say 21 canal reaches, but Table S1 shows 20 (presumably because 34 was measured
twice). Seems like 20 reaches is the correct one. This also applies to the “field sampling” section of
methods.

There was a canal missing in Table S1 (canal 47) in the original manuscript, 21 was correct. Table S1
now reports 34 canal reaches with CH₄ concentration and isotope data needed to estimate the percent
oxidized and diffusive fluxes, as 13 new sites were added during revisions.

There is a 35th site (canal 59) in Table S1 that we do not have canal water CH₄ concentration or δ¹³C-
CH₄ for due to measurement error. We do have a chamber flux measurement from that canal so we
included it in the supplemental summary table.

L88. “Across incubated waters, 53.5 ± 26.0% of the initial CH₄ was consumed over the incubation period
(17.6%-99.7%, Fig. 2A).”

This is interesting, and the following lines give info on general conc changes, but it would be nice to see a
multi-panel figure in the SI showing raw conc changes for each incubation.

To address this suggestion and a similar comment from Reviewer 3, we have added a supplementary
table (Table S2) that has the initial and final CH₄ concentrations and δ¹³C-CH₄ for incubated waters. We
have also revised Figure 2, combining percent concentration and isotopic ratio changes into 1 panel to
more clearly show the relationship between these changes:

**Figure 2. Methane consumption and resulting stable isotope fractionation in incubated canal waters.** A. Across incubated
waters, δ¹³C-CH₄ increased as the percent of initial CH₄ consumed increased. Each data point shows the mean change over ~50
394 hours of incubation ± standard error of replicates (Table S2). B. Histogram of α_{ox} values calculated from incubation data.

L96. “averaged”. Also L140, 178, L229. Be clear if these are means (presumably) or medians.
We have revised this line, and others noted here, to specify this value is the mean (L101):
*“Mean α_{ox} was 1.022 ± 0.009 across the incubated canal waters”*
L137. “It is unlikely that CH₄ concentration in canal waters is dictated only by the amount of CH₄
originally transported into canals from the surrounding landscape.”
Agreed, but in-situ production could also be relevant. Three decades (!) ago Roulet and Moore
considered the potential importance of lateral transport vs in-situ production
<https://cdnscepub.com/doi/abs/10.1139/x95-055>
We added an acknowledgement of the potential role of sediment and water column CH₄ production in the
following sections:
L149-154:
*“It is unlikely that CH₄ concentration in canal waters is dictated only by the amount of CH₄ originally*
*transported into canals from the surrounding landscape, including CH₄ produced in peat soils and canal*
*sediments. Methane produced in ombrotrophic tropical peat soils is highly depleted in ¹³C^{22,47}. Unlike in*
*lakes where $\delta^{13}\text{C-CH}_4$ in littoral sediments and adjacent groundwater can differ by more than 10‰⁴⁸ ,*
*porewater $\delta^{13}\text{C-CH}_4$ has not been shown to differ between canal bottoms and adjacent peat soils²¹.”*
L159-163:
*“Methane production in the water column could also influence canal water CH₄ concentration and $\delta^{13}\text{C-}$*
*CH₄. However, this is unlikely to explain our results because we did not observe net CH₄ production in*
*any of the laboratory incubations of canal waters, as CH₄ concentration decreased and $\delta^{13}\text{C-CH}_4$*
*increased in all incubated waters (Fig. 2A, Table S2).”*
L161. There are 35 data points for 21 canal reaches (and presumably the same for Fig 4a too). It isn’t
obvious where this 35 comes from. The “incubations” section mentions sample duplicates but I don’t think
the “canal CH₄” section does. Also, if you’re plotting sample duplicates as independent data points isn’t
this essentially pseudoreplication which will artificially inflate your sample size and therefore statistical
power? It seems more honest to calculate these correlations on means of the duplicates.
We re-ran statistical analyses and replotted figures using the means for canals with replicate samples.
The statistics and data visualization presented in the revised manuscript are based on the mean values
for each canal to avoid pseudoreplication. As such, each data point in all scatter plots represents a
separate canal reach.
As CH₄ concentration and isotopic composition may vary along a canal reach, summary statistics (means,
ranges, etc.) reported in the revised paper are based on all data - including spatial replicates within the
same reach- to report the full range of observations. We include data files of both the mean values and all
spatial replicates in the files uploaded to our data repository.
L189. “As such, shallow canal water depths likely support more CH₄ oxidation.”
This is interesting. Other studies, including of ditches (e.g.
<https://link.springer.com/article/10.1007/s13157-011-0170-y>) have found that CH₄ emissions are lower in
deeper ditches, because there is more scope for oxidation within the water column. There’s also ebullition
to consider, which is likely to be higher in shallow canals (because sediments will become warmer in
these systems).
We have added further discussion about the relationship between water depth and CH₄ in L229-246:
*“Given that higher dissolved oxygen and the presence of aquatic vegetation were observed in canals with*
*a shallower water depth (Fig. S6), canal water depth may indirectly mediate CH₄ oxidation in drainage*

canal waters. Overall, dissolved oxygen in the surface water of canals (0-10 cm) decreased with the
depth of water present in the canal (Kendall's $\tau = -0.41$, $p < 0.05$, Fig. S6). Dissolved CH₄ concentration,
and therefore estimated diffusive emissions, also had a weak but significant positive correlation with canal
water depth ($\tau = 0.26$, $p = 0.03$, Table S4). This result contradicts previous findings in drainage ditches in
temperate peatlands where CH₄ emissions had a weak negative correlation with depth⁵⁷, but these
differing results may be explained by how well canal waters are mixed and aerated. For example, while
we observed CH₄ oxidation in canals where dissolved oxygen is low ($< 2.5 \text{ mg L}^{-1}$) at the surface,
dissolved oxygen may become depleted at depth^{28,44} to below the concentration needed for aerobic
methanotrophs with high oxygen affinity. As such, CH₄ oxidation may be limited to the surface waters of
deeper canals, while in shallower canals oxidation may occur throughout the water column. Our study
also only explicitly considered diffusive emissions. Measurements of CH₄ ebullition from canals could
further clarify the role of water depth in shaping net canal CH₄ emissions, as ebullitive emissions vary with
water depth⁵⁸. Altogether, our results suggest that shallower, vegetated canals may attenuate a higher
percentage of CH₄ emissions through CH₄ oxidation.”

L208. “our finding that the majority of CH₄ transported into drainage canals is oxidized rather than emitted
is likely robust across peatland drainage canals in Southeast Asia.”

I acknowledge that it's always nice to claim that research findings are widely applicable but this is, to me,
overreaching. You measured 20 canals, once, in two study regions. I would suggest toning down this
sentence.

We have revised this sentence (L257-259) to state:

*“As such, we anticipate that water column CH₄ oxidation is prevalent across canals draining degraded
peatlands in Southeast Asia.”*

L218. Somewhere in the paper, and here seems a good place for it, it would be good to give some mean
fluxes from other tropical ditch studies for comparison.

We have added text (L269-277) and a supplementary table (Table 8) comparing our flux measurements
to past observations:

*“Our observations of canal CH₄ emissions estimated from dissolved CH₄ concentration ($72.2 \pm 151.2 \text{ mg}$
$\text{CH}_4 \text{ m}^{-2} \text{ d}^{-1}$) and collected using floating chambers ($94.9 \pm 142.3 \text{ mg CH}_4 \text{ m}^{-2} \text{ d}^{-1}$) are within range of past
observations from Indonesia^{26,27,63} and Malaysia²⁵ where mean emissions range from 2.8 to 1073 mg CH₄
$\text{m}^{-2} \text{ d}^{-1}$ (Table S8). The IPCC CH₄ Emissions Factor for canals in tropical peatlands of $618.9 \text{ mg CH}_4 \text{ m}^{-2}$
d^{-1} ($2259 \text{ kg CH}_4 \text{ ha}^{-1} \text{ y}^{-1}$) was based on the only reported data²⁶ at the time of the 2013 Wetlands
Supplement⁶⁴ This emission factor now represents the high end of field estimates to date among a still
small number of existing studies and should be reconsidered to more accurately inventory the
anthropogenic (e.g., from land use change) component⁶⁵ of CH₄ emissions from degraded tropical
peatlands.”*

L276. This says 100cm was your max water depth, Table S1 says 71cm.

We corrected this error. With the new sites, the max water depth at the location of sampling was 92 cm.

L277. Please add canal widths to Table S1.

We have added canal widths to Table S1.

L278. “and canals situated on peatlands under a variety of land uses (smallholder mixed
agriculture, smallholder oil palm, and industrial oil palm) to capture the heterogeneity of
drainage canals in the region”

Although you only have three in industrial oil palm. Perhaps worth emphasising that your study is biased
towards smallholder systems?

The additional data added to the revised paper include additional canals from industrial oil palm plantation
and canals from open undeveloped (e.g., deforested, no active land use) areas that were not represented
in the original manuscript. To be clear about land use representation in the study we added the following
text to L340-344:

*“Smallholder mixed agriculture is the most represented land use in this study, but the sampled canals*
*also include areas in smallholder plantations (pineapple and oil palm), industrial oil palm plantations, and*
*open undeveloped land, as well as 1 canal in a degraded forest, to capture the heterogeneity of drainage*
*canals in the region.”*

L298. If you're calculating k solely using windspeed I guess that means these canals aren't flowing (i.e.
there is zero turbulence). If so, best to explicitly say so in the text. But I admit to getting a little hesitant
when using wind speed to calculate emissions in these small waterbodies. These relationships between
wind speed and k have been tested in lakes, but can you be sure they are appropriate for small, sheltered
ditches (e.g. Fig. 1d)? I would suggest:

The canals have very low flow. Flow measured at a subset (n = 8) of canals was $0.12 \pm 0.03 \text{ m s}^{-1}$. We
have added this information in L405 and L427.

1. Including your k values somewhere so the reader can check them. How do they compare to k values
from other inland waters and from other ditches (or small waterbodies, e.g. ponds)?

2. Can you calculate some k values from your floating chamber deployments? Do they compare to your
windspeed-derived k values well?

In response to reviewer feedback, we have revised our approach to estimating k values. The revised
manuscript reports diffusive fluxes estimated using chamber-derived k values (L417-431):

*“We calculated gas transfer velocity (k , m d^{-1}) using data from the subset of canals where paired floating*
*chamber CH_4 fluxes and canal water CH_4 concentrations were collected using Eqn. (2):*

$$\text{Flux} = k(\text{CH}_{4\text{-canal}} - \text{CH}_{4\text{-eq}}) \quad (\text{Eqn. 2})$$

*Where $\text{CH}_{4\text{-canal}}$ is the concentration of CH_4 in canal water, $\text{CH}_{4\text{-eq}}$ is the CH_4 concentration at equilibrium*
*the atmosphere ($\text{CH}_{4\text{-eq}}$), and flux is the rate of CH_4 emissions measured using the floating chamber. We*
*used the median k value from the floating chamber deployments to estimate diffusive fluxes across all*
*sampled ($n = 34$) canals. While applying a uniform value introduces uncertainty into the estimates of*
*diffusive fluxes, conditions across the study region are characterized by high canal water temperature,*
*low canal flow velocity ($\sim 0.1 \text{ m s}^{-1}$), and low windspeed. As such, factors that strongly influence CH_4*
*degassing (e.g., solubility and turbulence) should have minimal variation relative to the ~ 600 -fold variation*
*in canal water CH_4 concentration across study sites. Values were normalized to k_{600} for literature*
*comparison. See Supplementary Section 1 for further discussion of approaches to estimate k .”*

Supplementary Text 1 includes a comparison of chamber- and wind speed-derived k values from our
sites to estimates to those from other shallow tropical waters, as well as those from forested ponds
determined via tracer experiments. Please see the response regarding k values to Reviewer 3 for further
discussion of these revisions.

There is some data in Fig. S4 where it seems like chambers are giving higher fluxes compared to k
values. Is this due to problems with the k method, or are chambers capturing bubble events too? Do you
seen any evidence of ebullition in these ditches?

We do see some evidence of ebullition in the canals which could influence this result. We are working to
quantify the ebullitive component of canal CH₄ emissions for a follow up study. Our revised approach to
estimating diffusive fluxes using chamber-derived k values brings fluxes from the two methods into closer
agreement (see Figure S5 and Table S8). We also identified a minor error in unit conversions in our
estimated flux calculations, that once corrected also increased the estimated fluxes alongside the change
in k value.

L307. "We collected canal waters at a subset (n = 13) of the drainage canals for incubation experiments"
Which canals? Can you add this into to Table S1.

We have added a * by canal ID numbers indicating which canals were included in the incubations in table
S1. Table S2 also reports incubation results for each canal, with location listed.

L309. How did you collect the deeper water samples? Please state.

We added the following text (L375-377):

*"Surface waters (~5 cm) were collected for all canals included in the incubation experiments, and at 5 of*
*the canals we collected water from ~10 cm above the canal bottom using gas-tight tubing and a hand*
*pump."*

L315. "Duplicate samples for each canal (and depth, if applicable) were acidified every ~24 hours to pH <
2 using 1.5M HCl to stop CH₄ oxidation."

So you had two replicates for each measurement and then (presumably) took the mean of both? This is
good, but it would be nice to see the reps data; how consistent are they to one another? Considering your
small sample size this info would be useful.

We have added a supplementary table (Table S2) that has the initial and final CH₄ concentrations and
δ¹³C-CH₄ (mean ± standard error) and the incubation time for all incubated waters.

L338. What depth pore water? Please state.

We have information about the depth of porewater sampling to L348-348. Porewater data is now also
reported in Table S3.

*"To measure the isotopic composition of source CH₄, we collected porewater profiles at 6 locations*
*adjacent to a subset of the sampled canals. As shallow porewater is the primary source of discharge to*
*drainage canals²¹, porewater was collected from 4-5 depths between 40 cm and 150 cm pending water*
*table depth.."*

Reviewer #3 (Remarks to the Author):

We thank Reviewer 3 for their comprehensive and constructive feedback. Here we summarize revisions
made in consideration of their major comments. Detailed responses to all comments follow below.

1. In consideration of Reviewer 3's concerns raised about the use of the boundary layer method
(and comments about flux calculations from the other 2 reviewers), we have revised our approach
to estimating CH₄ fluxes. In the revised manuscript we report fluxes calculated using gas transfer
velocity determined through the floating chamber deployments. We added additional

methodological information about our approach to estimating fluxes in the main text (L417-431)
and in Supplementary Text 1. We discuss these revisions in detail under Reviewer 3's comments
about flux estimation below.

- 2. We have added discussion about the potential influence of CH₄ production across the landscape
(e.g., in peat soils L151-159 and L219-225; in canal sediments and waters L149-163 and L217-
228) throughout the manuscript in response to questions raised by Reviewer 3 about the source
$\delta^{13}\text{C-CH}_4$ used in the Rayleigh model for estimating the fraction of CH₄ oxidized. We also added
more detail about the porewater $\delta^{13}\text{C-CH}_4$ data used in our calculations in the results/discussion
(L154-159) and methods sections (L348-355), as well as Table S3. For a variety of reasons
discussed at length below, we elected to not revise the source $\delta^{13}\text{C-CH}_4$ value used in our
calculations. In the revised manuscript we report the uncertainty introduced by using a uniform
value for source $\delta^{13}\text{C-CH}_4$ (L127-129) and include a new supplementary figure (Fig. S3) showing
how varying source $\delta^{13}\text{C-CH}_4$ impacts our estimate of the fraction of CH₄ oxidized vs. emitted.
- 3. Reviewer 3 raised important questions about seasonality, as in climates with large seasonal
variation in temperature (e.g., northern and temperate regions) and rainfall (e.g., monsoonal
climates) there can be large variation in $\delta^{13}\text{C-CH}_4$ due to changes in the rate and/or pathway of
CH₄ production. The climate in our study region is equatorial with hot, humid, and heavy to very
heavy rainfall all year; as such we do not anticipate large seasonal variation in $\delta^{13}\text{C-CH}_4$. Variable
temperature, dissolved oxygen, etc. can also impact the isotopic fractionation factor of oxidation,
another critical parameter in our approach to estimating the proportion of the fraction of CH₄
oxidized. We have added additional discussion of potential controls of the isotopic fractionation
we observed in incubations (L106-113). In short, we did not find any significant environmental
correlates of this isotopic fractionation. Like for source $\delta^{13}\text{C-CH}_4$, we report the uncertainty
introduced to our estimates of oxidation due to the variability in isotopic fractionation in the main
text (L125-127) and Figure S3.

Please find detailed responses to Reviewer 3's comments, including further discussion of the points
raised above, below. We appreciate their feedback on the manuscript and feel the revisions inspired by
their comments aided us in producing a more comprehensive assessment of our findings.

The manuscript by Perryman et al. is an original and valuable contribution that extends the knowledge on
CH₄ dynamics with particular focus on the role and controls of CH₄ oxidation in mitigating CH₄ emissions
from Southeast Asia's tropical peatland drainage canals. The manuscript's main and most relevant finding
is that CH₄ oxidation is an important regulator of CH₄ emissions also in tropical peatland drainage
canals. This is reportedly the first study to document isotopic fractionation due to aerobic CH₄ oxidation in
these environments, which is crucial for enhancing our understanding of CH₄ oxidation's isotopic
fractionation factor in freshwater aquatic settings.

While the conclusions and claims are generally supported by the results, the portrayal of certain controls
over CH₄ oxidation as being significant appears somewhat overstated, considering the weak to moderate
correlations shown in the figures.

We have revised these statements to better reflect the strength of the observed correlations, for example:

*L194-196: "Of the studied controls on CH₄ oxidation, dissolved oxygen and aquatic vegetation had the*
*most significant influence on the percent of CH₄ oxidized in canals as determined by canal water $\delta^{13}\text{C-}$*
*CH₄."*

The methods are commonly used in the field and there are enough information for the work to be
reproduced. However, some choices may increase the uncertainty of the results and their limitations are

not discussed or well-motivated. The main points here are 1) the choice of estimating CH₄ fluxes using
the boundary layer method; 2) the source $\delta^{13}\text{C-CH}_4$ used in the Rayleigh model for estimating the
fraction of CH₄ oxidation; and 3) results not representative of potential seasonal variability of the source
$\delta^{13}\text{C-CH}_4$ used in calculations and actual seasonality in CH₄ oxidation. Detailed comments about these
points can be found below, and in the specific comments.

The choice for estimating CH₄ fluxes using the boundary layer method instead of their flux measurements
using floating chambers, which could also be used to calculate site and gas (CH₄) specific k₆₀₀. Flux
estimates would be more robust if using the actual flux measurements and k₆₀₀ derived from your
measurements instead of wind models developed for CO₂ emissions from lakes.

Please see our detailed response below (lines 982-1026) regarding this matter. In the revised manuscript,
we estimate diffusive CH₄ emissions using gas transfer velocities estimated from chamber deployments
instead of from wind speed (methodological information: L417-431, Supplementary Text 1). This
increases our estimates of diffusive emissions, but they are still within range of the few previous estimates
of canal CH₄ from other major peatland areas in Indonesia and Malaysia (see Table S8).

Another important point that deserves attention is the use of groundwater adjacent to the canal as the
isotopic signature source in the Rayleigh model. CH₄ production may also take place in the canal's
sediment, and this production may have a heavier isotopic signature attributed to a different CH₄
production pathway that could lead to overestimation of results. In addition, the input of CH₄ to the water
column may be a mixture of CH₄ produced in the canal's sediment and from groundwater through lateral
flow. The latter may vary seasonally, changing the isotopic signature of the source CH₄ influencing the
estimates of CH₄ oxidation over seasons.

Please see our detailed response below (line 1069-1191 of this document) about the selection of $\delta^{13}\text{C-CH}_4$ -
CH₄ source value. In the revised manuscript, we have included a more thorough discussion of these
uncertainties. To capture the uncertainty introduced, we report how our estimate varies in response to
changing the source signature in L127-129:

*“Similarly, considering the standard deviation of the porewater source $\delta^{13}\text{C-CH}_4$ measurements, the mean
666 percent oxidized could range from $68.2 \pm 16.1\%$ to $82.4 \pm 8.9\%$ (Fig. S3).”*

We also added discussion of the possible influence of sediment production and/or different methanogenic
pathways to the text (L151-159, 217-228). We appreciate the points raised by reviewer 3, but ultimately
we did not elect to change the source value we used in our calculation of the percent oxidized, but
instead focused on quantifying and discussing the associated uncertainties.

Regarding seasonality, the climate in the study region is equatorial (e.g., hot, humid, and heavy to very
heavy rainfall all year). There is not a markedly wetter or drier season like in monsoonal tropical climates,
but July and August have slightly lower total monthly precipitation (~150-200 mm/month vs. 200-300
676 mm/month for the rest of the year). Temperatures are consistently warm year-round. This is in contrast to
677 northern or temperate regions where seasonal variation in temperature and primary production can have
678 a large influence on the rate and/or pathway of CH₄ production. Extremely limited process-oriented data
on CH₄ cycling in degraded tropical peatlands (e.g., measurements of isotopes, incubations, etc.) exist,
limiting our understanding of potential seasonal variation in these processes. Prior work from a less
heavily degraded site in Brunei that also lacks strong seasonality suggests that the amount of CH₄
advection from peat soils (due to variability in porewater CH₄ concentration and porewater discharge)
may vary across the year, but porewater $\delta^{13}\text{C-CH}_4$ remains depleted year-round (~ -75‰; Somers, Hoyt,
et al., 2023; *JGR Biogeosciences*).

To consider the representativeness of our results given the lack of temporal data, we compare our results
in Supplementary Tables 7 and 8 (dissolved CH₄, dissolved oxygen, and CH₄ fluxes) to other studies of
canals in regions of Southeast Asia with more pronounced precipitation seasonality. As other studies do
not report δ¹³C-CH₄, we cannot contextualize our isotopic results. This emphasizes the need for further
work in this region.

Seasonality may also affect CH₄ oxidation rates and fluxes, yet the work does not cover or discuss
limitations about seasonality. The authors should be careful with extrapolations that may not be
representative for whole year.

We removed the extrapolation to rates of CH₄ oxidation per year in the conclusion. Please see our
discussion above in response to the points raised above about seasonality. We acknowledge the lack of
seasonal representation in the text in L254-257, and also show that relevant data (Table S7; dissolved
oxygen and dissolved CH₄) are within range of past observations collected in areas during pronounced
wet or dry seasons:

*“While our study was not conducted during pronounced wet or dry periods, the dissolved CH₄ and oxygen*
*concentrations measured in our study fall within the range observed across Southeast Asia under varying*
*land uses and seasons^{21,27,44,60,61} (Table S7).”*

Other points to consider:

- ● Some relevant papers about CH₄ oxidation based on stable isotopes in tropical aquatic
environments may be relevant in the introduction and/or to further explore in the discussions (e.g.
Barbosa et al. 20181; Sawakuchi et al 20162; Tyler et al. 19973; Zhang et al. 20134).
 - ○ We have added suggested references where appropriate in the revised
introduction/discussion sections (ref. 29, 31, and 43):
 - ○ L54-56 :*“One process that strongly influences freshwater CH₄ emissions is microbial*
*oxidation of CH₄ to carbon dioxide. In other tropical freshwaters (e.g., rivers, lakes) CH₄*
*oxidation attenuates CH₄ emissions by 40 to nearly 100%^{22,29-31}.”*
 - ○ L102-105: *“The range of α_{ox} encompasses...results from incubations of soil from*
*subtropical rice paddies⁴³ (α_{ox} of 1.025-1.033) that are often used in estimates of CH₄*
*oxidation in tropical freshwaters^{29,31}.”*
- ● Further exploration and discussion of the observed variability of isotopic fractionation would be
relevant since this is one of the most unique results reported. Isotopic fractionation values
estimated for rice paddy environments (Tyler et al. 1997; Zhang et al. 2013) could add some
insights and perhaps is a more similar and relevant type of environment to compare with than
northern lakes.
 - ○ See responses below in lines 804-823 of this document related to this point. We have
added discussion of the potential drivers of variability in isotopic fractionation and
comparisons to the aforementioned studies conducted in warmer climates to L98-113.
- ● Limited number of replicates in incubations might hide potential analytical errors or local variability
of oxidation rates and isotope fractionation in the incubation experiment.
 - ○ We acknowledge that we have limited replicates. We are limited in the amount of sample
material we can bring back for analysis in the USA due to baggage weight restrictions
when departing Indonesia. We report variability between replicate incubations in Fig. 2A
and Table S2. Overall, the change in dissolved CH₄ and δ¹³C-CH₄ over the incubation
period was larger than variability between replicate samples.
- ● Using mean values for the source δ¹³C-CH₄ from groundwater instead of site-specific values
may obscure potential variability between sites that influence the analysis of the controlling

factors. I would suggest using mean values only for sites where you did not measure groundwater
$\delta^{13}\text{C-CH}_4$.

- ○ We discuss this point and revisions made in consideration of it thoroughly in lines 1069-
1191 of this document. In brief:
- ○ We appreciate the suggested revision to our calculations, but ultimately we elected not to
revise the source $\delta^{13}\text{C-CH}_4$ for a number of reasons. From our porewater observations,
$\delta^{13}\text{C-CH}_4$ varied more by depth (up to 20‰ variability between samples from the same
profile) than across the landscape. Furthermore, the canals drain porewater from along
their entire reach; thus a point measurement from one location along the total length of
the canal does not necessarily represent the $\delta^{13}\text{C}$ of the whole mass of CH_4 advected
into the canal. Given these considerations, and others discussed at length in lines 1069-
1191 of this document, we elected to not revise our approach to selecting source $\delta^{13}\text{C-CH}_4$.
Instead, we focused our revisions on better quantifying and describing the
uncertainty introduced by using a uniform source $\delta^{13}\text{C-CH}_4$ and discussing factors that
may influence our results (e.g., production pathways and sources of CH_4)
- ○ We acknowledge that using a mean value for source $\delta^{13}\text{C-CH}_4$ may not accurately
account for variability between sites and introduce additional uncertainty. We have
revised the manuscript to explicitly report the uncertainty introduced by using a mean
source $\delta^{13}\text{C-CH}_4$ value in the main text (L125-129) and Fig. S3.
- ● Other relevant papers may also be useful for discussing the isotopic signature of source methane
used as reference for the oxidation calculations (Thottathil and Prairie 2021; Schenk et al. 2021).
- ○ We included a reference suggested above in L152-154:
*“Unlike in lakes where $\delta^{13}\text{C-CH}_4$ in littoral sediments and adjacent groundwater can differ
by more than 10‰⁴⁸, porewater $\delta^{13}\text{C-CH}_4$ has not been shown to differ between canal
bottoms and adjacent peat soils²¹.”*

Specific comments

L70-74. Perhaps it is unnecessary and unusual to summarize the main results at the end of the
introduction.

We agree this practice varies across journals. However, we elected to summarize key findings at the end
of the introduction following the example of other papers in this journal. See examples in recent papers on
tropical peatlands published in Nature Communications: Cooper et al. (2020)
[<https://doi.org/10.1038/s41467-020-14298-w>] and Hodgkins et al. (2018)
[<https://doi.org/10.1038/s41467-018-06050-2>].

L85. The title of the subsection seems more a method's subsection title.

We have revised the title of this subsection to *“CH₄ consumption and isotopic fractionation observed
during incubations”*.

L91-93. CH_4 oxidation in incubations may be higher when starting CH_4 concentrations are higher and it
would be good to describe discuss the influence of starting concentrations on the results. Extra
supplementary table or figure showing the start and end CH_4 concentration and $\delta^{13}\text{C-CH}_4$ and the time
incubated would be valuable to understand the changes you show in Figure 2.

Yes, potential CH_4 oxidation rates were strongly influenced by the initial CH_4 concentration; however
initial CH_4 concentration did not affect the isotopic fractionation observed via incubations. As measuring
the isotopic fractionation of CH_4 oxidation was the primary aim of the incubations, initial CH_4
concentration did not influence downstream results (i.e., estimates of percent oxidized in situ made using
fractionation factors measured in vitro). To address this point, we have added the following text to L105-

107:

“While CH_4 oxidation rates varied with initial CH_4 concentration, we did not observe a correlation between
α_{ox} and initial CH_4 concentration, nor α_{ox} and CH_4 oxidation rate (Fig. S2).”

We also added panels to Figure S2. to show the lack of relationship between initial CH_4 concentration and
isotopic fractionation, as well as oxidation rate and isotopic fractionation:

**Figure S2.** A. CH_4 oxidation rates from incubations of canal waters varied with initial CH_4 concentration. B-C. Isotopic fractionation
(α_{ox}) did not vary with initial CH_4 concentration nor CH_4 oxidation rate.

To address the later portion of this comment, we have added a supplementary table (Table S2) that has
the initial and final CH_4 concentrations and $\delta^{13}\text{C}-\text{CH}_4$ and the incubation time for all incubated waters.

L95-97. Further description and discussion of the observed variability would be important for the field to
improve understanding of isotopic fractionation if the variability observed could be associated with some
environmental factors.

L99-103. Would be nice to discuss the variability of isotopic fractionation in more detail. Thottathil et al
2022 cited here show that the fractionation varied with depth and temperature, and although the overall
range be similar that may be local factors controlling the variability of isotopic fractionation that need to be
explored and discussed. Also, the numbers you present from references 37 and 38 are not the correct
overall range reported in these papers combined.

We have revised the second paragraph (L98-113) of this subsection to include discussion about the
variability of isotopic fractionation. Thank you for catching the error in the numbers from ref. 37-38; (now
ref. 41-42). In the revised text, we now compare the whole range we observed for α_{ox} (1.002-1.039) in
comparison to that observed by ref. 41-42 (1.0184-1.0208 and 1.004-1.038, respectively).

“From these data we calculated the first empirically derived isotopic fractionation factors for CH_4
oxidation³⁹ (α_{ox}) in peat-draining freshwaters. Ecosystem-specific values for α_{ox} are critical to estimating
the percent of CH_4 that is oxidized rather than emitted from the natural environment^{40,41}. Mean α_{ox} was
1.022 ± 0.009 across the incubated canal waters (range: 1.002-1.039; Fig. 2B). The range of α_{ox}
encompasses past observations from northern and temperate freshwaters incubated under in situ
dissolved CH_4 and oxygen concentrations and temperature^{41,42}, as well as results from incubations of soil
from subtropical rice paddies⁴³ (α_{ox} of 1.025-1.033) that are often used in estimates of CH_4 oxidation in
tropical freshwaters^{29,31}. While CH_4 oxidation rates varied with initial CH_4 concentration, we did not
observe a correlation between α_{ox} and initial CH_4 concentration, nor α_{ox} and CH_4 oxidation rate (Fig. S2).
Recent work in temperate lakes identified temperature, pH, and dissolved O_2 as potential controls on
α_{ox} ⁴². Of these factors, α_{ox} was only weakly positively correlated with the initial dissolved O_2 present in

*each of the incubated waters ($p = 0.07$). α_{ox} did not vary between surface and bottom waters of the subset*
*of canals sampled at two depths for incubation experiments (1.024 ± 0.006 vs. 1.023 ± 0.012 , $n = 4$*
*canals). As we did not find significant environmental correlates of α_{ox} , we used the mean value to*
*estimate in situ CH_4 oxidation as discussed below.”*

L116. The range in “(range: 47.5-91.4%; Fig. 3A)” is not something really evident to see in Figure 3A.
We omitted the figure reference during revisions.

L131. Here you mention the exponential decrease in Fig 3B, but in the figure you show a linear
relationship and even mention a linear relationship in the caption. I see you use the log scale in the y-axis
but maybe good to make it clearer to the reader that will not find an exponential relationship in the figure.
We omitted the word “exponentially” from this sentence to avoid confusion.

L133-134. How the information from the reference “positive correlation between gene markers for
methanotrophic bacteria and $\delta^{13}C-CH_4$ ”, relates to the negative relationship you have observed?
We revised these sentences to make the connection between these observations more clear (L142-148):

*“Previous observations in tropical river networks³¹ also observed a negative relationship between the*
*concentration of CH_4 in river waters and $\delta^{13}C-CH_4$. In these rivers $\delta^{13}C-CH_4$ also had a positive*
*relationship with gene markers for methanotrophic bacteria, indicating that variation in CH_4 concentration*
*and $\delta^{13}C-CH_4$ is influenced by CH_4 oxidation. The consistent relationship between CH_4 concentration and*
*$\delta^{13}C-CH_4$ observed across the drainage canals in our study and these tropical rivers supports the idea*
*that differences in dissolved CH_4 concentrations between canal reaches are influenced by CH_4 oxidation.”*

L140-144. Not really, you already have a large variation in porewater $\delta^{13}C-CH_4$ and this variation could
increase if all sites were measured. I would also expect CH_4 production in the canal and that this
production would have a less negative $\delta^{13}C-CH_4$ because of acetoclastic methanogenesis. Using only
$\delta^{13}C-CH_4$ from porewater outside the canal might lead to overestimation of results. See more comments
on this below (comments for L336-338).
We have included further discussion about the variation of porewater $\delta^{13}C-CH_4$ and potential contributions
from in-canal production in L149-168. See comments below (lines 1127-1154 of this document) for
discussion about the potential role of acetoclastic methanogenesis.

*“It is unlikely that CH_4 concentration in canal waters is dictated only by the amount of CH_4 originally*
*transported into canals from the surrounding landscape, including CH_4 produced in peat soils and canal*
*sediments. Methane produced in ombrotrophic tropical peat soils is highly depleted in ^{13}C ^{22,47}. Unlike in*
*lakes where $\delta^{13}C-CH_4$ in littoral sediments and adjacent groundwater can differ by more than 10‰⁴⁸,*
*porewater $\delta^{13}C-CH_4$ has not been shown to differ between canal bottoms and adjacent peat soils²¹.*
*Porewater $\delta^{13}C-CH_4$ collected from 6 profiles (40 to 150 cm depth) located alongside canal waters in our*
*study region had a mean $\delta^{13}C-CH_4$ of $-85.0 \pm 5.9\%$, which was consistently more depleted than any*
*observed canal $\delta^{13}C-CH_4$ value (Table S1, S3). Porewater $\delta^{13}C-CH_4$ varied more between sample depths*
*within each profile than between profiles collected across the landscape, suggesting source $\delta^{13}C-CH_4$ is*
*similarly depleted in ^{13}C throughout the study region. Methane production in the water column could also*
*influence canal water CH_4 concentration and $\delta^{13}C-CH_4$. However, this is unlikely to explain our results*
*because we did not observe net CH_4 production in any of the laboratory incubations of canal waters, as*
*CH_4 concentration decreased and $\delta^{13}C-CH_4$ increased in all incubated waters (Fig. 2A, Table S2). If canal*
*water CH_4 concentration were influenced solely by the total amount of CH_4 produced and then*
*transported into canal waters, we would expect canal water $\delta^{13}C-CH_4$ to be similarly depleted across*
*canals and not vary systematically with dissolved CH_4 concentration. Given that CH_4 concentrations*

varied ~600-fold alongside a ~40‰ range in $\delta^{13}\text{C-CH}_4$, our results indicate that CH_4 oxidation has a
significant influence on canal water CH_4 concentration and $\delta^{13}\text{C-CH}_4$.”

L150. Confusing Fig 3C show linear relationship.

We omitted the word “exponentially” from this sentence to avoid confusion.

Figure 3. In this figure, panels B and C basically show the same information (as shown by the points
distribution) and this is because diffusive fluxes are dependent on CH_4 concentrations and % CH_4
oxidized dependent on $\delta^{13}\text{C-CH}_4$. That makes it logical to see a similar pattern and does not necessarily
show that diffusive fluxes are related to oxidation. It would have been interesting to add an extra panel
showing the relationship between measured fluxes (floating chambers) and oxidation rates from
incubations, which would be independent of concentrations and $\delta^{13}\text{C-CH}_4$.

To limit redundancy in the presentation of the data in the main text, we have revised Figure 3 to show
violin plots of CH_4 and $\delta^{13}\text{C-CH}_4$ to show the distribution of these data without having a redundant
relationship shown in the same figure. The panel showing the relationship between CH_4 and $\delta^{13}\text{C-CH}_4$
has been moved to the supplement (Fig. S4).

**Figure 3. Survey of drainage canal CH_4 concentrations and $\delta^{13}\text{C-CH}_4$ reveal the impact of CH_4 oxidation on canal CH_4**
**emissions.** A. Curve showing the relationship between canal water $\delta^{13}\text{C-CH}_4$ and estimated percent CH_4 oxidized across the mean
(black line) and ± 1 standard deviation (shaded region) of the laboratory derived α_{ox} value. B-C. Surface water $\delta^{13}\text{C-CH}_4$ and
dissolved CH_4 concentration across the studied canals ($n = 34$). D. Estimates of the percent of CH_4 oxidized versus estimated
diffusive CH_4 flux across the studied canals. For panels B-D each dot represents a canal. The shaded region of panel D represents
the 95% confidence interval associated with the linear relationship. Dissolved CH_4 concentration and estimated diffusive CH_4 flux
are shown on a \log_{10} scale in panels C and D.

Since we did not normalize the concentration of CH_4 in the incubated canal waters (e.g., no spike with
CH_4 standard in vial headspace), the oxidation rates from the incubations are highly dependent on initial
CH_4 concentration, as discussed above. The canals that had the highest potential CH_4 oxidation rates
from incubations are canals with high initial dissolved CH_4 concentrations and more depleted initial $\delta^{13}\text{C-CH}_4$,
indicating that less oxidation occurs in the field. Canals with lower potential oxidation rates
determined in incubations had less initial dissolved CH_4 and more enriched initial $\delta^{13}\text{C-CH}_4$, suggesting

more oxidation occurs in situ in those waters. As such, comparing chamber CH₄ emissions to incubation
CH₄ oxidation rates would not accurately reflect the relationship between oxidation and CH₄ emissions.

L172. The %CH₄ oxidized and dissolved O₂ relationship shown in Fig 4A is not strong and in Fig 4B you
only show that %CH₄ oxidized in different between open water and vegetation, and not a relationship. I
suggest reformulating and toning down the statement that O₂ and vegetation strongly influence the %
oxidation.

We have revised the opening of this section to state (L194-196):

*“Of the studied controls on CH₄ oxidation, dissolved oxygen and aquatic vegetation had the most*
*significant influence on the percent of CH₄ oxidized in canals as determined by canal water δ¹³C-CH₄.”*

L177-182. I wonder if this difference could in some extent be attributed to a heavier source of δ¹³C-CH₄
produced by acetoclastic methanogenesis, which is more likely to happen in vegetated areas.

We added discussion about the potential for vegetation to influence methanogenesis pathways in L217-
228:

*“The deposition of labile organic matter from vegetation could also stimulate acetoclastic*
*methanogenesis, which like CH₄ oxidation would contribute towards larger δ¹³C-CH₄ in vegetated*
*canals³⁸. However, acetoclastic methanogenesis likely contributes little to the δ¹³C-CH₄ in vegetated*
*canals because hydrogenotrophic methanogenesis has been identified as the dominant pathway in the*
*ombrotrophic tropical peatlands of Southeast Asia²² and the Americas^{47,55}. Disturbance in peatlands in*
*Southeast Asia has been observed to increase the abundance of plant functional types associated with*
*acetoclastic methanogenesis, like graminoids, but this shift does not appear to increase the abundance of*
*acetoclastic methanogens⁵⁶. While we cannot rule out the possible influence of acetoclastic*
*methanogenesis on canal water δ¹³C-CH₄, the lower dissolved CH₄ concentration in vegetated canals (p*
*= 0.02, Table S5) lends more support to the idea that vegetation enhances CH₄ oxidation rather than*
*acetoclastic CH₄ production in canals.”*

L186-189. Here too, you describe a clear/strong relationship between O₂ and depth that is not very
evident in Fig S5.

We omitted the phrase “because dissolved oxygen and vegetation cover closely followed shifts in canal
water depth” to tone down the strength of this finding and better reflect the potential impact of canal water
depth. This paragraph now begins with the following statement (L229-231):

*“Given that higher dissolved oxygen and the presence of aquatic vegetation were observed in canals with*
*a shallower water depth (Fig. S6), canal water depth may indirectly mediate CH₄ oxidation in drainage*
*canal waters.”*

L230-231. In Fig S7 you show those two outliers marked in the grey box. Please explain what the reason
for that could be.

These data points are from downstream of a canal block, as stated in the figure caption. The canal block
may influence turbulence and/or mixing, causing very recently advected (and therefore more isotopically
depleted) CH₄ to be degassed. Our collaborators have also seen local residents and/or farmers using this
particular canal reach for washing tools and equipment. This canal (#31) was one of the deeper (68 cm)
and more isotopically depleted (canal water δ¹³C-CH₄ = -65.2 ± 0.1‰), but it had relatively low dissolved
CH₄ (0.76 ± 0.02 μM), deviating from the overall trends observed in the larger dataset. Our hypothesis is
that turbulence from the canal block or recent use of the canal for washing may have degassed CH₄ from

the canal water prior to our sample collection. As transport across the air-water interface results in little to
no isotopic fractionation, the remnant CH₄ would be depleted in ¹³C (e.g., closer to source value) but CH₄
concentration would be low.

L276. Please inform what season is that and what that season means for the lateral water and CH₄ input
to canals.

This region of Indonesia does not have pronounced wet or dry seasons. We have clarified this in L334-
338:

*“Drainage canals in lowland peatlands were sampled in Kubu Raya and Mempawah Districts, West*
*Kalimantan, Indonesia. Canals were sampled in Kubu Raya in May 2023 and Mempawah in April 2024.*
*This region has an equatorial rainfall pattern with no clear wet and dry season⁷⁰. There is heavy rainfall*
*year-round, but the driest months of the year usually occur in July or August.”*

L278-280. Describe how they differ and how it should affect lateral water flow, CH₄ in groundwater, canal
depth, and presence of vegetation.

There were no discernable patterns in canal depth, CH₄ in canal water or porewater (concentration or
¹³C), or vegetation across land uses (Table S6). Our results do not indicate strong land use patterns
across canals, but perhaps a large sampling or synthesis effort could reveal any systematic variation
using a larger dataset.

L296: Please describe in more details how k₆₀₀ was calculated including the assumptions and model
parameters from Cole and Caraco 5. I wonder how reliable these estimates are considering that the Cole
and Caraco 5 model was created for lakes and CO₂. Here, I imagine two sources of uncertainty, 1) lakes
have larger open areas for wind and gusts to develop in comparison to canals that are more sheltered
and canals may be a lotic environment where water velocity could also influence k and is not accounted
by the Cole and Caraco model, and 2) the model was created for CO₂ and recent research, Pajala et al.
6 has observed a higher k for CO₂ compared with CH₄, meaning that the calculated k for CH₄ using this
model could overestimate k for CH₄ and consequently the fluxes estimates. Why not use the k from the
floating chambers you have deployed and the mean for sites without floating chamber measurements?
This would give you more robust and site-specific k estimates.

We have now fully revised our approach to estimating diffusive fluxes, using chamber derived k values,
which is discussed in more detail below. In response to the reviewer’s specific comments:

1) Flow velocity is very low in these canals. Flow measured at a subset (n = 8) of canals was 0.12 ±
0.03 m s⁻¹.

2) Estimates of k from the chamber measurements (from 12 of 34 sampled canals) are ~2x larger
on average than k values estimated using wind speed from the same canal. As such, it is unlikely
that the flux estimates in the original manuscript were overestimates. The estimated fluxes in the
original manuscript were low compared to previous floating chamber measurements of CH₄
emissions from canals draining peatlands in other regions of Indonesia and Malaysia.

More generally, in the revised manuscript, we estimate diffusive fluxes using chamber-derived k values.
The distribution of k estimates from the floating chambers is highly skewed, and in instances where
replicate chamber measurements were made (n = 4 canals) there was considerable variability in k within
the same canal reach. For example, for the same canal reach we estimate k of 0.3 to 3.1 m d⁻¹. Given the
large inter- and intra-canal variability in k estimated from chambers, in the revised manuscript we estimate
diffusive fluxes across all sampled canals using the median k value (k₆₀₀ = 1.15 m d⁻¹) from chamber

measurements. We acknowledge that in reality k will not be uniform across sites. However, given the low
wind (often $< 0.5 \text{ m s}^{-1}$) and low flow conditions across canals, as well as their relative similarity in size
and sheltering due to deforestation in the study region, variation in canal water CH_4 concentration across
canals should be the main factor driving variability in diffusive CH_4 emissions.

Revising our approach does increase our estimated fluxes, but they remain within range of those
previously observed from drainage canals in other major peatland regions of Indonesia and Malaysia. We
have added a supplementary table comparing our estimated diffusive fluxes as well as our chamber
fluxes to previous observations (Table S8). Supplementary Text 1 also provides discussion of approaches
to estimating k , how they impact our results, and comparison of k values from our study and others in
similar environments.

We have revised the methods text to reflect our revised approach in L417-431:

*“We calculated gas transfer velocity (k , m d^{-1}) using data from the subset of canals where paired floating
chamber CH_4 fluxes and canal water CH_4 concentrations were collected using Eqn. (2):*

$$\text{Flux} = k(\text{CH}_{4\text{-canal}} - \text{CH}_{4\text{-eq}}) \quad (\text{Eqn. 2})$$

*Where $\text{CH}_{4\text{-canal}}$ is the concentration of CH_4 in canal water, $\text{CH}_{4\text{-eq}}$ is the CH_4 concentration at equilibrium
the atmosphere ($\text{CH}_{4\text{-eq}}$), and flux is the rate of CH_4 emissions measured using the floating chamber. We
used the median k value from the floating chamber deployments to estimate diffusive fluxes across all
sampled ($n = 34$) canals. While applying a uniform value introduces uncertainty into the estimates of
diffusive fluxes, conditions across the study region are characterized by high canal water temperature,
low canal flow velocity ($\sim 0.1 \text{ m s}^{-1}$), and low windspeed. As such, factors that strongly influence CH_4
degassing (e.g., solubility and turbulence) should have minimal variation relative to the ~ 600 -fold variation
in canal water CH_4 concentration across study sites. Values were normalized to k_{600} for literature
comparison. See Supplementary Section 1 for further discussion of approaches to estimate k .”*

L308. Specify the depth of surface water collected for incubations.

L309. The deeper water collected from some canals (40-70 cm) was related to a specific % of the total
depth or distance from sediment (e.g. 80 % of the canal depth)?

We added the depth ($\sim 5 \text{ cm}$) of surface water collected for incubations in L375. The sampling depth for
deeper samples was not related to a specific depth/distance, but rather as deep as we could sample
without disturbing the soft peat sediment underlying the canal to minimize disturbance for other ongoing
measurements at our study sites. In general, this was $\sim 10 \text{ cm}$ from the canal bottom. We revised L375-
377 to be more clear about the sampling approach for incubated waters:

*“Surface waters ($\sim 5 \text{ cm}$) were collected for all canals included in the incubation experiments, and at 5 of
the canals we collected water from $\sim 10 \text{ cm}$ above the canal bottom using gas-tight tubing and a hand
pump.”*

L310. Typo. “pending”.

Typo corrected during revision.

L315. My experience with incubations for CH_4 oxidation is that it can have large variability between
replicates, and it would be good to show this variability. Since you only have two replicates what does not
make standard deviation very meaningful, you could show the range in Figure 2 or in a supplementary
table with more information about the incubations as mentioned above.

We have added a supplementary table that has the initial and final CH₄ concentrations and δ¹³C-CH₄
(mean ± standard error) and the incubation time for all incubated waters (Table S2). The revised Figure
2A now also shows the standard error of replicate samples for the percent CH₄ consumed and change in
δ¹³C-CH₄ over the incubation period.

L327. You mean Eqn 2?

Equation numbers changed during revision due to reorganization of the methods text.

L336-338. Unclear if you have used a mean δ¹³C-CH₄ as the δ_{source} for all sites in the calculation. The
standard deviation shows a considerable difference in δ¹³C-CH₄ between sites or replicates and if a
mean was used this would add errors to the results of single sites. Please consider describing in the
methods what values did you use for the calculations and discuss how this limits the results, especially for
sites where you did measure porewater. I also wonder about the potential and large variability between
δ¹³C-CH₄ in porewater outside the canals and in the canal's sediment. Recent studies 7,8 show large
variation of δ¹³C-CH₄ in bubbles released from lake sediments attributed to different pathways of CH₄
production that could also be the case for these canals. Not all CH₄ may come from groundwater/lateral
flow and CH₄ production may occur in the canal, especially canal with vegetation, where fresh organic
matter is available and less negative δ¹³C-CH₄ would be expected because of acetoclastic
methanogenesis. Using different and more negative δ¹³C-CH₄ could largely overestimate the fraction of
CH₄ oxidation, and this should be thoroughly discussed.

We are sensitive to the many potential sources of uncertainty and/or variability of the source δ¹³C-CH₄
value. The revised manuscript thoroughly considers these, including additional quantification and
visualization of the uncertainty introduced into our results by the source δ¹³C-CH₄ value (#1 below),
discussion of the variability of the porewater δ¹³C-CH₄ measurements (#2), methods clarification (#3), and
text discussing the potential influence of varying methanogenic pathways (#4) and CH₄ production in
canal sediments and/or waters (#5).

**1. We acknowledge the variability of our porewater δ¹³C-CH₄ measurements introduces**
**uncertainty into our estimates of the percent oxidized.** The revised manuscript quantifies and
visualizes the uncertainty introduced by the variation in porewater δ¹³C-CH₄ in L127-129 and Fig.S3:
“Similarly, considering the standard deviation of the porewater source δ¹³C-CH₄ measurements, the mean
1080 percent oxidized could range from 68.2 ± 16.1% to 82.4 ± 8.9% (Fig. S3).”

**Figure S3.** A. Density plot showing estimates of the percent of CH₄ oxidized in canal waters using the mean or ± one standard
deviation value of our estimate of the isotopic fractionation of oxidation (1.022 ± 0.008). B. Density plot showing estimates of the
1084 percent of CH₄ oxidized in canal waters using the mean or ± one standard deviation value of our estimate of the source δ¹³C-CH₄,
(85.0 ± 5.9‰).

**2. We have added text that describes the variability in porewater $\delta^{13}\text{C-CH}_4$ and clarifies our**
**porewater sampling scheme:**

L154-159: *“Porewater $\delta^{13}\text{C-CH}_4$ collected from 6 profiles (40 to 150 cm depth) located alongside canal*
*waters in our study region had a mean $\delta^{13}\text{C-CH}_4$ of $-85.0 \pm 5.9\%$, which was consistently more depleted*
*than any observed canal $\delta^{13}\text{C-CH}_4$ value (Table S1, S3). Porewater $\delta^{13}\text{C-CH}_4$ varied more between*
*sample depths within each profile than between profiles collected across the landscape, suggesting*
*source $\delta^{13}\text{C-CH}_4$ is similarly depleted in ^{13}C throughout the study region.”*

For example, in two of the six profiles there is a 15-20‰ increase in $\delta^{13}\text{C-CH}_4$ from the bottom to the top
of the profile, with the most significant increase between the upper 2 samples (top sample ~5 cm below
water table). The increase in porewater $\delta^{13}\text{C-CH}_4$ towards the peat surface coincides with a decrease in
CH_4 concentration. This suggests the trend is primarily driven by oxidation and not a shift in
methanogenic pathway. As such, the more shallow porewater with heavier $\delta^{13}\text{C-CH}_4$ contributes much
less to the total CH_4 pool transported into canals than the deeper porewater with more depleted $\delta^{13}\text{C-CH}_4$
and higher CH_4 concentration. Weighted by concentration - and therefore contribution to the canal CH_4
pool - the mean porewater $\delta^{13}\text{C-CH}_4$ is -84.92% (compared to the arithmetic mean of -84.98%).

We also clarified our porewater sampling approach in L348-351:

*“To measure the isotopic composition of source CH_4 , we collected porewater profiles at 6 locations*
*adjacent to a subset of the sampled canals. As shallow porewater is the primary source of discharge to*
*drainage canals²¹, porewater was collected from 4-5 depths between 40 cm and 150 cm pending water*
*table depth.”*

**3. We appreciate the reviewer's idea to use site-specific $\delta^{13}\text{C-CH}_4$ when available. However,**
**unfortunately this approach is unable to meaningfully reduce uncertainty in our case.** Canals
drain porewater along their entire length (which can encompass peat soils under a variety of land
uses and/or vegetation). As such, using a value from a single point is likely a poorer representation of
the bulk $\delta^{13}\text{C-CH}_4$ transported into the canal than using an average value representing a larger area.
Instead of revising this portion of our data analysis, we chose to focus our revisions around more
rigorously quantifying uncertainties and discussing factors that could influence source $\delta^{13}\text{C-CH}_4$
value. We clarified the source $\delta^{13}\text{C-CH}_4$ value used in our calculations in L447-451:

*“The results presented in the main analyses and figures are estimates of the percent oxidized based*
*on mean observed values of α_{ox} (1.022 ± 0.009 , from incubations) and δ_{source} ($-85.0 \pm 5.9\%$, $n=27$*
*measurements from 6 porewater profiles). To characterize the uncertainty of our estimates due to*
*variability in α_{ox} and δ_{source} , we also report how our estimate varies when using ± 1 standard deviation*
*of α_{ox} or δ_{source} in Eqn. 3.”*

**4. While acetoclastic methanogenesis can play an important role in minerotrophic tropical**
**peatlands (e.g. Buessecker et al., 2021; *Frontiers in Microbiology*), we anticipate that**
**hydrogenotrophic methanogenesis is the dominant methanogenic pathway across the highly**
**disturbed, ombrotrophic peatlands we studied for the following reasons:**

a. Other ombrotrophic peatlands in Borneo, Peru, and Panama (Buessecker et al., 2021; Holmes et
al., 2015, *Global Biogeochemical Cycles*; Hoyt, 2017, *unpublished PhD thesis*) have very
depleted porewater $\delta^{13}\text{C-CH}_4$ reflecting hydrogenotrophic methanogenesis like observed in our
study. Paired $\delta^{13}\text{C-CO}_2$ measurements from these other studies further indicate hydrogenotrophy
is the dominant methanogenic pathway in these ombrotrophic peatlands.

- b. Tropical peatland disturbance influences the quality of DOM (Gandois et al., 2013,
*Biogeochemistry*; Gandois et al., 2014, *Geochimica et Cosmochimica Acta*) as more DOM is
derived from decomposed peat carbon vs. fresh plant inputs. As such, disturbed peatlands have
less labile DOM for acetoclastic methanogenesis.

Vegetation in canals could influence CH₄ production by contributing fresh DOM that could stimulate
acetoclastic methanogenesis. We address this point in L217-228:

*“The deposition of labile organic matter from vegetation could also stimulate acetoclastic*
*methanogenesis, which like CH₄ oxidation would contribute towards larger δ¹³C-CH₄ in vegetated*
*canals³⁸. However, acetoclastic methanogenesis likely contributes little to the δ¹³C-CH₄ in vegetated*
*canals because hydrogenotrophic methanogenesis has been identified as the dominant pathway in the*
*ombrotrophic tropical peatlands of Southeast Asia²² and the Americas^{47,55}. Disturbance in peatlands in*
*Southeast Asia has been observed to increase the abundance of plant functional types associated with*
*acetoclastic methanogenesis, like graminoids, but this shift does not appear to increase the abundance of*
*acetoclastic methanogens⁵⁶. While we cannot rule out the possible influence of acetoclastic*
*methanogenesis on canal water δ¹³C-CH₄, the lower dissolved CH₄ concentration in vegetated canals (p*
*= 0.02, Table S5) lends more support to the idea that vegetation enhances CH₄ oxidation rather than*
*acetoclastic CH₄ production in canals.”*

**5. We acknowledge that CH₄ production may occur in canal sediments and/or canal waters, but**
**do not anticipate this is a large and/or different source of CH₄ to canals for the following**
**reasons:**

- a. We did not observe net CH₄ production in any of the canal water incubation experiments.
b. Work from our collaborators in a peatland drainage canal in northern Borneo (Somers et al.,
2023; *JGR Biogeosciences*) found that δ¹³C-CH₄ in peat underlying the canal was -69.2 ± 4.8‰,
while porewater down to 2.5 m (interval where most CH₄ advected to the canal originates) in the
~60 m on either side of the canal (within our sample scheme) had δ¹³C-CH₄ of -71.7 ± 9.0‰ to -
68.4 ± 5.1‰, showing these CH₄ sources are isotopically similar.
c. Overall, the total area of the canal bottom is much smaller than the total area of peatland drained
by a given canal. Therefore, the proportional contribution of production in underlying sediments to
CH₄ in canal waters is much lower than the CH₄ transported from the peat. If CH₄ produced in
canal sediments had a distinct δ¹³C-CH₄, the signature of the peat porewater still would dominate
due to its larger contribution to the CH₄ pool.

We address the potential influence of in-canal (sediment or water column) production on source δ¹³C-CH₄
in L149-168:

*“It is unlikely that CH₄ concentration in canal waters is dictated only by the amount of CH₄ originally*
*transported into canals from the surrounding landscape, including CH₄ produced in peat soils and canal*
*sediments. Methane produced in ombrotrophic tropical peat soils is highly depleted in ¹³C^{22,47}. Unlike in*
*lakes where δ¹³C-CH₄ in littoral sediments and adjacent groundwater can differ by more than 10‰⁴⁸, ,*
*porewater δ¹³C-CH₄ has not been shown to differ between canal bottoms and adjacent peat soils²¹.*
*Porewater δ¹³C-CH₄ collected from 6 profiles (40 to 150 cm depth) located alongside canal waters in our*
*study region had a mean δ¹³C-CH₄ of -85.0 ± 5.9‰, which was consistently more depleted than any*
*observed canal δ¹³C-CH₄ value (Table S1, S3). Porewater δ¹³C-CH₄ varied more between sample depths*
*within each profile than between profiles collected across the landscape, suggesting source δ¹³C-CH₄ is*
*similarly depleted in ¹³C throughout the study region. Methane production in the water column could also*
*influence canal water CH₄ concentration and δ¹³C-CH₄. However, this is unlikely to explain our results*

*because we did not observe net CH₄ production in any of the laboratory incubations of canal waters, as*
*CH₄ concentration decreased and δ¹³C-CH₄ increased in all incubated waters (Fig. 2A, Table S2). If canal*
*water CH₄ concentration were influenced solely by the total amount of CH₄ produced and then*
*transported into canal waters, we would expect canal water δ¹³C-CH₄ to be similarly depleted across*
*canals and not vary systematically with dissolved CH₄ concentration. Given that CH₄ concentrations*
*varied ~600-fold alongside a ~40‰ range in δ¹³C-CH₄, our results indicate that CH₄ oxidation has a*
*significant influence on canal water CH₄ concentration and δ¹³C-CH₄.*"

Additionally, porewater results are not available on Table S1, and it would be nice to see the porewater's
δ¹³C-CH₄ and concentration from each site where it was collected.

We have included a new supplementary table (Table S3) with the porewater data.

L339. Please briefly describe the piezometer. Does it have a membrane or porous material that could
create isotopic fractionation? Also indicate the model of the piezometer together with the company's
name.

The portable piezometer has a coarse polypropylene screen to prevent debris from clogging the tubing.
We do not anticipate that this screen creates isotopic fractionation. We added the following description of
the piezometer (L351-354):

*"Porewater was collected using a portable piezometer made of 3/8" stainless steel tubing housing 1/4"*
*polyethylene tubing equipped with a coarse polypropylene screen to prevent collection of coarse debris*
*(SedPoints, M.H.E. Products)."*

L352. "In the main text we report..." please add the section where this is reported. I could not find any
explanation about this in the main text.

We have revised this sentence to say (L439):

*"Oxidation efficiency (f_{ox}) was calculated using a Rayleigh model for closed systems."*

*f_{ox} is the fraction oxidized. In the paper we report percent oxidized (f_{ox} * 100), as stated in L444-445.*

L364. In this section, please add more information about the acceptable R² from the linear regression to
accept a flux measurement. Here it would also be nice to describe how much water flows in the canals
and if the chambers were allowed to follow the flow or if kept on the same position, and if so, who this
could influence the flux measurement by changing the water turbulence with the chamber. I also wonder
in shallow places if the edges of the chamber could hit the sediment influencing the measurements.

Please report the flux measurements results in Table S1.

We have added methodological details discussed here in L404-406 and L412-413:

*"The floating chamber was not held in place, but due to low canal water flow (stagnant to ~0.1 m s⁻¹) that*
*chamber did not travel during flux measurement."*

*"Fluxes were accepted if the linear increase in CH₄ over time met the standards of R² > 0.9 and p < 0.05."*

We added the flux measurements where collected to Table S1.

References

1. Barbosa, P. M. et al. High rates of methane oxidation in an Amazon floodplain lake.

Biogeochemistry 137, 351-365 (2018). <https://doi.org/10.1007/s10533-018-0425-2>

2. Sawakuchi, H. O. et al. Oxidative mitigation of aquatic methane emissions in large Amazonian
rivers. Global Change Biology 22, 1075-1085 (2016). <https://doi.org/10.1111/gcb.13169>

3. Tyler, S. C., Bilek, R. S., Sass, R. L. & Fisher, F. M. Methane oxidation and pathways of production in a Texas paddy field deduced from measurements of flux, delta C-13, and delta D of CH₄. *Global Biogeochemical Cycles* 11, 323-348 (1997). <https://doi.org/10.1029/97gb01624>
 4. Zhang, G. B. et al. Pathway of CH₄ production, fraction of CH₄ oxidized, and C-13 isotope fractionation in a straw-incorporated rice field. *Biogeosciences* 10, 3375-3389 (2013). <https://doi.org/10.5194/bg-10-3375-2013>
 5. Cole, J. J. & Caraco, N. F. Atmospheric exchange of carbon dioxide in a low-wind oligotrophic lake measured by the addition of SF₆. *Limnology and Oceanography* 43, 647-656 (1998).
 6. Pajala, G. et al. Higher Apparent Gas Transfer Velocities for CO₂ Compared to CH₄ in Small Lakes. *Environmental Science & Technology* 57, 8578-8587 (2023). <https://doi.org/10.1021/acs.est.2c09230>
 7. Schenk, J. et al. Methane in Lakes: Variability in Stable Carbon Isotopic Composition and the Potential Importance of Groundwater Input. *Frontiers in Earth Science* 9 (2021). <https://doi.org/10.3389/feart.2021.722215>
 8. Thottathil, S. D. & Prairie, Y. T. Coupling of stable carbon isotopic signature of methane and ebullitive fluxes in northern temperate lakes. *Science of The Total Environment* 777, 146117 (2021). <https://doi.org/https://doi.org/10.1016/j.scitotenv.2021.146117>

Author responses to new reviewer comments are in blue text.

Reviewer #1 (Remarks to the Author):

I am satisfied with the responses and changes made to the manuscript.

We thank Reviewer 1 for their time and constructive feedback which helped improve our manuscript.

Reviewer #2 (Remarks to the Author):

Overall, the authors have thoroughly considered my original comments and revised their manuscript accordingly. I certainly didn't require, or expect, additional data to appear, but the new data from 13 canal reaches in another area further strengthen the small dataset and are a welcome addition. I have two small comments on this new draft. Otherwise, I find the manuscript acceptable for publication and look forward to seeing the published version.

We thank Dr. Peacock for his constructive feedback on our work. His suggestions helped us strengthen the manuscript. Please see below for our responses to the 2 minor comments on the revised manuscript.

Original comment:

L315. "Duplicate samples for each canal (and depth, if applicable) were acidified every ~24 hours to pH <2 using 1.5M HCl to stop CH₄ oxidation."

So you had two replicates for each measurement and then (presumably) took the mean of both? This is good, but it would be nice to see the reps data; how consistent are they to one another? Considering your small sample size this info would be useful.

Author response:

We have added a supplementary table (Table S2) that has the initial and final CH₄ concentrations and δ¹³C-CH₄ (mean ± standard error) and the incubation time for all incubated waters.

New comment:

My original comment asked for the replicate sample data to be included – that is still hidden in Table S2 by the use of means (although the SEM values give hints). It isn't in the online data either: the file Canal_Water_Incubations_Perryman has the replicate measurements of dissolved CH₄ and d¹³C_{CH4} (or are these the replicate *changes* in these parameters?) but this isn't sufficient. To be clear, I would like to see the raw, replicate data, set out as in Table S2 (CH₄ T₀, CH₄ T_{final}, ¹³C T₀, ¹³C T_{final}) whereby each individual, replicate incubation has its own line (i.e. not averaged together) – unless these data are already hiding somewhere in the SI but if so I don't see it. It's potentially important/interesting for the reader to see how consistent your reps are.

The file "Canal_Water_Incubations_Perryman.xlsx" in the Zenodo repository for this manuscript (<https://doi.org/10.5281/zenodo.11155160>) has the data requested here. Each row reports the CH₄ concentration (in μM) and ¹³C (in ‰) for the 2 replicate vials for each canal at each time point. For example, the following table in the same format reports the results for canal #10:

Canal	Depth	Rep	Hours	dissolvedCH4	¹³ C
10	Surface	A	0.0	0.52	-50.5
10	Surface	B	0.0	0.47	-50.0
10	Surface	A	53.6	0.42	-47.9
10	Surface	B	53.6	0.37	-42.7

At T₀, CH₄ concentration of the 2 replicates (A and B) was 0.52 and 0.47 μM, and the ¹³C of those replicates -50.5 and -50.0‰. After 53.6 hours of incubation, the T_{final} CH₄ concentration and ¹³C of the 2 replicates was 0.42 and 0.37 μM and -47.9 and -42.7‰, respectively. Table S2 reports the mean ± standard error of the CH₄ concentration and ¹³C of the 2 replicates.

To be clear, for each incubation we collected 2 vials for T_0 (preserved in the field) and 2 for T_{final} (preserved after ~50 hours of incubation). Each vial was analyzed for CH_4 concentration and ^{13}C once. The mean of the 2 replicate vials for each time point were used in calculations and reported in Table S2, and the raw data from each time point is in the "Canal_Water_Incubations_Perryman.xlsx" on Zenodo. These data are also in the Source Data file for Figure 2a that will be uploaded alongside final paper revisions.

We have revised the description for Table S2 to direct readers to our Source Data File to view the individual (i.e., raw) data points:

“Table S2. Dissolved CH_4 concentration and $\delta^{13}\text{C}\text{-CH}_4$ across incubated canal waters. Values are mean \pm standard error of 2 replicate vials for each time point. Time indicates the incubation length before determination of the final CH_4 concentrations and $\delta^{13}\text{C}\text{-CH}_4$. For depth, S = surface and D = deep. Values for the CH_4 concentrations and $\delta^{13}\text{C}\text{-CH}_4$ of individual replicates are available in the manuscript Source Data file.”

One minor comment

L344. “Open undeveloped land” is slightly vague because to a casual reader it hides human action. Perhaps change to “deforested undeveloped land” (or similar)?

“Open undeveloped land” is a land use classification used in research on peatlands in Southeast Asia. Our use of “open undeveloped land” follows the definition of Miettinen et al. (2016, *Global Ecology and Conservation*) which includes deforested areas with ferns/low shrub (< 2m) vegetation and ‘clearance’ areas with no vegetation including recently burned areas.

We have revised this sentence to clarify the definition of this classification, but we did not revise the name of the group to be consistent with the literature on peatlands in this region (e.g., Bowen et al., 2024, *Nature Geoscience*; Dadap et al., 2021, *AGU Advances*; Deshmukh et al., 2020, *Global Change Biology*; Deshmukh et al., 2021, *Nature Geoscience*; Miettinen et al., 2017, *Environmental Management*; Miettinen et al., 2017, *Environmental Research Letters*).

L294-298: “Smallholder mixed agriculture is the most represented land use in this study, but the sampled canals also include areas in smallholder plantations (pineapple and oil palm), industrial oil palm plantations, and open undeveloped land (i.e., deforested and/or burned areas), as well as 1 canal in a degraded forest, to capture the heterogeneity of drainage canals in the region.”

Mike Peacock

Reviewer #3 (Remarks to the Author):

Comments on the reviewed manuscript NCOMMS-24-34376 “Fate of methane in canals draining tropical peatlands” by Perryman et al.

Perryman et al. have done an amazing job addressing my concerns in the revised manuscript. All the points raised were carefully considered in the revised text, including new relevant information and clarifications. I don’t have any further considerations about the manuscript and I believe this is a valuable contribution to the field.

Similarly to the manuscript, I would like to thank the authors for this well-structured and nice to read response letter.

We thank Reviewer 3 for their thorough feedback on the manuscript. The revisions made in response to their review strengthened the work and we appreciate the time and care taken to provide constructive feedback.

Comments on NCOMMS-24-34376 "Fate of methane in canals draining tropical peatlands" by Perryman et al.

The manuscript by Perryman et al. is an original and valuable contribution that extends the knowledge on CH₄ dynamics with particular focus on the role and controls of CH₄ oxidation in mitigating CH₄ emissions from Southeast Asia's tropical peatland drainage canals. The manuscript's main and most relevant finding is that CH₄ oxidation is an important regulator of CH₄ emissions also in tropical peatland drainage canals. This is reportedly the first study to document isotopic fractionation due to aerobic CH₄ oxidation in these environments, which is crucial for enhancing our understanding of CH₄ oxidation's isotopic fractionation factor in freshwater aquatic settings.

While the conclusions and claims are generally supported by the results, the portrayal of certain controls over CH₄ oxidation as being significant appears somewhat overstated, considering the weak to moderate correlations shown in the figures.

The methods are commonly used in the field and there are enough information for the work to be reproduced. However, some choices may increase the uncertainty of the results and their limitations are not discussed or well-motivated. The main points here are 1) the choice of estimating CH₄ fluxes using the boundary layer method; 2) the source $\delta^{13}\text{C-CH}_4$ used in the Rayleigh model for estimating the fraction of CH₄ oxidation; and 3) results not representative of potential seasonal variability of the source $\delta^{13}\text{C-CH}_4$ used in calculations and actual seasonality in CH₄ oxidation. Detailed comments about these points can be found below, and in the specific comments.

The choice for estimating CH₄ fluxes using the boundary layer method instead of their flux measurements using floating chambers, which could also be used to calculate site and gas (CH₄) specific k₆₀₀. Flux estimates would be more robust if using the actual flux measurements and k₆₀₀ derived from your measurements instead of wind models developed for CO₂ emissions from lakes.

Another important point that deserves attention is the use of groundwater adjacent to the canal as the isotopic signature source in the Rayleigh model. CH₄ production may also take place in the canal's sediment, and this production may have a heavier isotopic signature attributed to a different CH₄ production pathway that could lead to overestimation of results. In addition, the input of CH₄ to the water column may be a mixture of CH₄ produced in the canal's sediment and from groundwater through lateral flow. The latter may vary seasonally, changing the isotopic signature of the source CH₄ influencing the estimates of CH₄ oxidation over seasons.

Seasonality may also affect CH₄ oxidation rates and fluxes, yet the work does not cover or discuss limitations about seasonality. The authors should be careful with extrapolations that may not be representative for whole year.

Other points to consider:

Some relevant papers about CH₄ oxidation based on stable isotopes in tropical aquatic environments may be relevant in the introduction and/or to further explore in the discussions (e.g. Barbosa et al. 2018¹; Sawakuchi et al 2016²; Tyler et al. 1997³; Zhang et al. 2013⁴).

Further exploration and discussion of the observed variability of isotopic fractionation would be relevant since this is one of the most unique results reported. Isotopic fractionation values estimated for rice paddy environments (Tyler et al. 1997; Zhang et al. 2013) could add some insights and perhaps is a more similar and relevant type of environment to compare with than northern lakes.

Limited number of replicates in incubations might hide potential analytical errors or local variability of oxidation rates and isotope fractionation in the incubation experiment.

Using mean values for the source $\delta^{13}\text{C-CH}_4$ from groundwater instead of site-specific values may obscure potential variability between sites that influence the analysis of the controlling factors. I would suggest using mean values only for sites where you did not measure groundwater $\delta^{13}\text{C-CH}_4$.

Other relevant papers may also be useful for discussing the isotopic signature of source methane used as reference for the oxidation calculations (Thottathil and Prairie 2021; Schenk et al. 2021).

Specific comments

L70-74. Perhaps it is unnecessary and unusual to summarize the main results at the end of the introduction.

L85. The title of the subsection seems more a method's subsection title.

L91-93. CH_4 oxidation in incubations may be higher when starting CH_4 concentrations are higher and it would be good to describe discuss the influence of starting concentrations on the results. Extra supplementary table or figure showing the start and end CH_4 concentration and $\delta^{13}\text{C-CH}_4$ and the time incubated would be valuable to understand the changes you show in Figure 2.

L95-97. Further description and discussion of the observed variability would be important for the field to improve understanding of isotopic fractionation if the variability observed could be associated with some environmental factors.

L99-103. Would be nice to discuss the variability of isotopic fractionation in more detail. Thottathil et al 2022 cited here show that the fractionation varied with depth and temperature, and although the overall range be similar that may be local factors controlling the variability of isotopic fractionation that need to be explored and discussed. Also, the numbers you present from references 37 and 38 are not the correct overall range reported in these papers combined.

L116. The range in "(range: 47.5-91.4%; Fig. 3A)" is not something really evident to see in Figure 3A.

L131. Here you mention the exponential decrease in Fig 3B, but in the figure you show a linear relationship and even mention a linear relationship in the caption. I see you use the log scale in the y-axis but maybe good to make it clearer to the reader that will not find an exponential relationship in the figure.

L133-134. How the information from the reference “positive correlation between gene markers for methanotrophic bacteria and $\delta^{13}\text{C-CH}_4$ ”, relates to the negative relationship you have observed?

L140-144. Not really, you already have a large variation in porewater $\delta^{13}\text{C-CH}_4$ and this variation could increase if all sites were measured. I would also expect CH_4 production in the canal and that this production would have a less negative $\delta^{13}\text{C-CH}_4$ because of acetoclastic methanogenesis. Using only $\delta^{13}\text{C-CH}_4$ from porewater outside the canal might lead to overestimation of results. See more comments on this below (comments for L336-338).

L150. Confusing Fig 3C show linear relationship.

Figure 3. In this figure, panels B and C basically show the same information (as shown by the points distribution) and this is because diffusive fluxes are dependent on CH_4 concentrations and % CH_4 oxidized dependent on $\delta^{13}\text{C-CH}_4$. That makes it logical to see a similar pattern and does not necessarily show that diffusive fluxes are related to oxidation. It would have been interesting to add an extra panel showing the relationship between measured fluxes (floating chambers) and oxidation rates from incubations, which would be independent of concentrations and $\delta^{13}\text{C-CH}_4$.

L172. The % CH_4 oxidized and dissolved O_2 relationship shown in Fig 4A is not strong and in Fig 4B you only show that % CH_4 oxidized is different between open water and vegetation, and not a relationship. I suggest reformulating and toning down the statement that O_2 and vegetation strongly influence the % oxidation.

L177-182. I wonder if this difference could in some extent be attributed to a heavier source of $\delta^{13}\text{C-CH}_4$ produced by acetoclastic methanogenesis, which is more likely to happen in vegetated areas.

L186-189. Here too, you describe a clear/strong relationship between O_2 and depth that is not very evident in Fig S5.

L230-231. In Fig S7 you show those two outliers marked in the grey box. Please explain what the reason for that could be.

L276. Please inform what season is that and what that season means for the lateral water and CH₄ input to canals.

L278-280. Describe how they differ and how it should affect lateral water flow, CH₄ in groundwater, canal depth, and presence of vegetation.

L296: Please describe in more details how k_{600} was calculated including the assumptions and model parameters from Cole and Caraco⁵. I wonder how reliable these estimates are considering that the Cole and Caraco⁵ model was created for lakes and CO₂. Here, I imagine two sources of uncertainty, 1) lakes have larger open areas for wind and gusts to develop in comparison to canals that are more sheltered and canals may be a lotic environment where water velocity could also influence k and is not accounted by the Cole and Caraco model, and 2) the model was created for CO₂ and recent research, Pajala et al.⁶ has observed a higher k for CO₂ compared with CH₄, meaning that the calculated k for CH₄ using this model could overestimate k for CH₄ and consequently the fluxes estimates. Why not use the k from the floating chambers you have deployed and the mean for sites without floating chamber measurements? This would give you more robust and site-specific k estimates.

L308. Specify the depth of surface water collected for incubations.

L309. The deeper water collected from some canals (40-70 cm) was related to a specific % of the total depth or distance from sediment (e.g. 80 % of the canal depth)?

L310. Typo. "pending".

L315. My experience with incubations for CH₄ oxidation is that it can have large variability between replicates, and it would be good to show this variability. Since you only have two replicates what does not make standard deviation very meaningful, you could show the range in Figure 2 or in a supplementary table with more information about the incubations as mentioned above.

L327. You mean Eqn 2?

L336-338. Unclear if you have used a mean $\delta^{13}\text{C-CH}_4$ as the δ_{source} for all sites in the calculation. The standard deviation shows a considerable difference in $\delta^{13}\text{C-CH}_4$ between sites or replicates and if a mean was used this would add errors to the results of single sites. Please consider describing in the methods what values did you use for the calculations and discuss how this limits the results, especially for sites where you did measure porewater. I also wonder about the potential and large variability between $\delta^{13}\text{C-CH}_4$ in porewater outside

the canals and in the canal's sediment. Recent studies ^{7,8} show large variation of $\delta^{13}\text{C-CH}_4$ in bubbles released from lake sediments attributed to different pathways of CH_4 production that could also be the case for these canals. Not all CH_4 may come from groundwater/lateral flow and CH_4 production may occur in the canal, especially canal with vegetation, where fresh organic matter is available and less negative $\delta^{13}\text{C-CH}_4$ would be expected because of acetoclastic methanogenesis. Using different and more negative $\delta^{13}\text{C-CH}_4$ could largely overestimate the fraction of CH_4 oxidation, and this should be thoroughly discussed.

Additionally, porewater results are not available on Table S1, and it would be nice to see the porewater's $\delta^{13}\text{C-CH}_4$ and concentration from each site where it was collected.

L339. Please briefly describe the piezometer. Does it have a membrane or porous material that could create isotopic fractionation? Also indicate the model of the piezometer together with the company's name.

L352. "In the main text we report..." please add the section where this is reported. I could not find any explanation about this in the main text.

L364. In this section, please add more information about the acceptable R^2 from the linear regression to accept a flux measurement. Here it would also be nice to describe how much water flows in the canals and if the chambers were allowed to follow the flow or if kept on the same position, and if so, who this could influence the flux measurement by changing the water turbulence with the chamber. I also wonder in shallow places if the edges of the chamber could hit the sediment influencing the measurements. Please report the flux measurements results in Table S1.

References

- Barbosa, P. M. *et al.* High rates of methane oxidation in an Amazon floodplain lake. *Biogeochemistry* **137**, 351-365 (2018). <https://doi.org/10.1007/s10533-018-0425-2>
- Sawakuchi, H. O. *et al.* Oxidative mitigation of aquatic methane emissions in large Amazonian rivers. *Global Change Biology* **22**, 1075-1085 (2016). <https://doi.org/10.1111/gcb.13169>
- Tyler, S. C., Bilek, R. S., Sass, R. L. & Fisher, F. M. Methane oxidation and pathways of production in a Texas paddy field deduced from measurements of flux, delta C-13, and delta D of CH_4 . *Global Biogeochemical Cycles* **11**, 323-348 (1997). <https://doi.org/10.1029/97gb01624>
- Zhang, G. B. *et al.* Pathway of CH_4 production, fraction of CH_4 oxidized, and C-13 isotope fractionation in a straw-incorporated rice field. *Biogeosciences* **10**, 3375-3389 (2013). <https://doi.org/10.5194/bg-10-3375-2013>
- Cole, J. J. & Caraco, N. F. Atmospheric exchange of carbon dioxide in a low-wind oligotrophic lake measured by the addition of SF_6 . *Limnology and Oceanography* **43**, 647-656 (1998).
- Pajala, G. *et al.* Higher Apparent Gas Transfer Velocities for CO_2 Compared to CH_4 in Small Lakes. *Environmental Science & Technology* **57**, 8578-8587 (2023). <https://doi.org/10.1021/acs.est.2c09230>

- Schenk, J. *et al.* Methane in Lakes: Variability in Stable Carbon Isotopic Composition and the Potential Importance of Groundwater Input. *Frontiers in Earth Science* **9** (2021). <https://doi.org/10.3389/feart.2021.722215>
- Thottathil, S. D. & Prairie, Y. T. Coupling of stable carbon isotopic signature of methane and ebullitive fluxes in northern temperate lakes. *Science of The Total Environment* **777**, 146117 (2021). <https://doi.org/https://doi.org/10.1016/j.scitotenv.2021.146117>